# Temporal dynamics of CH$_4$ emission pathways in the subsaline reed wetland of Lake Neusiedl

Pamela Alessandra Baur[1,2], Thiago Rodrigues-Oliveira[3], Karin Hager[3], Zhen-Hao Luo[3], Christa Schleper[2,3], and Stephan Glatzel[1,2]

[1]University of Vienna, Faculty of Earth Sciences, Geography and Astronomy, Department of Geography and Regional Research, Geoecology, Josef-Holaubek-Platz 2, 1090 Vienna, Austria.
[2]University of Vienna, Faculty of Life Sciences, Vienna Doctoral School of Ecology and Evolution (VDSEE), Djerassiplatz 1, 1030 Vienna, Austria.
[3]University of Vienna, Faculty of Life Sciences, Functional and Evolutionary Ecology, Djerassiplatz 1, 1030 Vienna, Austria.

**Correspondence:** Pamela Alessandra Baur (pamela.baur@univie.ac.at)

**Abstract.** Wetlands are a natural source of methane (CH$_4$) emissions and represent a substantial uncertainty in the global CH$_4$ budget. Furthermore, wetlands dominated by reed (*Phragmites australis*) have various CH$_4$ emission pathways, some of which are challenging to quantify (e.g., ebullition) or require additional research (e.g., plant-mediated transport). Plant-mediated transport is often not considered in greenhouse gas balance models for wetlands, nor is the correct mode of gas transport in reeds (pressurized flow). Therefore, further field studies on CH$_4$ emissions in wetlands, especially reed wetlands, are needed to reduce uncertainties in the global CH$_4$ budget and to improve the parametrization and implementation of emission pathways in greenhouse gas balance models of wetlands. This field study investigates all assessable CH$_4$ emission pathways and interfaces (diffusion, ebullition, plant-mediated transport) with various chamber types over four seasons and over the entire diel cycle (24 h) in the subsaline and dynamic reed wetland of Lake Neusiedl in Austria. The pathways of CH$_4$ formation (methanogenesis) were examined in each season by determining $\delta^{13}$C source signatures, and over the course of a year, by investigating specific microbial groups (methanogens, methanotrophs, and sulfate reducers) in the sediments. The highest CH$_4$ emissions were observed in summer, regardless of the emission pathway, with the highest emissions in all seasons occurring via the plant-mediated transport. Significant differences in CH$_4$ fluxes were observed between the plant-mediated transport and diffusion pathway in each season. However, a distinct diel cycle of CH$_4$ flux was exclusively observed via plant-mediated transport during summer. The source signatures $\delta^{13}$C-CH$_4$ exhibit seasonal variation, with the highest $^{13}$C-depletion occurring in fall. Despite the different seasonal source signatures, the dominant methanogenic pathway remains acetoclastic throughout all seasons. Desiccation of the reed ecosystem resulted in a reduction in methanogenic microbial diversity in the sediments over the course of one year. Concurrently, the drought resulted in an increase and dominance of oxygen-tolerant *Methanomicrobiales*.

## 1 Introduction

Methanogens are well known for being producers of methane (CH$_4$) in various environments, with wetlands constituting the main natural source of this gas on the planet (Wang et al., 1996). However, wetland CH$_4$ emissions (102–200 Tg CH$_4$ a$^{-1}$,

2008–2017) represent a significant source of uncertainty in the global $CH_4$ budget (Saunois et al., 2020). In wetlands, methanogenesis predominantly occurs in anoxic sediments and can be classified into three different methanogenic pathways based on the substrate utilized for microbial anaerobic $CH_4$ production (Conrad, 2020a): acetoclastic (acetate), hydrogenotrophic ($H_2/CO_2$), and methylotrophic (methyl compounds, e.g., methanol). The stable carbon isotope ratio of $CH_4$ ($\delta^{13}$C-$CH_4$) is useful for distinguishing between biogenic and thermogenic sources of $CH_4$ production, with thermogenic sources generally showing more enriched $^{13}$C values between $-20$ and $-50$ ‰ (Whiticar, 1999). The ratio can be also used to ascertain the dominant methanogenic pathway and the influence of $CH_4$ oxidation in different environments (Whiticar et al., 1986; Whiticar, 1999; Conrad, 2005). Acetoclastic methanogenesis produces $\delta^{13}$C-$CH_4$ values between $-50$ and $-65$ ‰ in sediments, whereas $CH_4$ produced by hydrogenotrophic methanogenesis is more depleted in $^{13}$C, with values ranging from $-60$ to $-100$ ‰ (Whiticar et al., 1986). The presence of specific methanogens in sediments can also provide valuable insights into wetlands. For example, only two genera, *Methanosaeta* and *Methanosarcina*, conduct acetoclastic methanogenesis, whereas hydrogenotrophic methanogenesis is more common across methanogens (Megonigal et al., 2003). Additionally, some methanogens specialize in growing on one, two, or multiple substrates (Megonigal et al., 2003).

Methanogenesis is particularly relevant when considering that it is the final step of organic matter degradation in anoxic ecosystems (Schlesinger and Bernhardt, 2013). However, due to its lower energy yield, methanogenesis occurs last, with other alternative electron acceptors (nitrate > manganese > iron > sulfate) being used first as substrates for anaerobic respiration. Consequently, the concentration of these substrates and the presence of specific microbial communities in sediments, aside from methanogens, such as sulfate reducers or methanotrophs, can serve as indicators of environmental conditions. These indicators offer insights into the intricate interactions, processes, and methanogenic pathways that occur within and beyond the sediments (Hilderbrand et al., 2020; Soman et al., 2024; Zhang et al., 2020). Wetlands with sulfate-containing sediments usually occur along the coast or are influenced by seawater (brackish). Sulfate-reducing bacteria convert sulfate ($SO_4^{2-}$) into less oxidized sulfur compounds, such as hydrogen sulfide ($H_2S$), to derive energy. Due to their higher energy yield, sulfate-reducing bacteria often outcompete methanogens for substrate uptake (Lovley and Klug, 1983; Lovley et al., 1982; Lovley and Goodwin, 1988; King, 1984), resulting in reduced or suppressed $CH_4$ production. However, there is a sulfate-methane transition zone in some environments, where they co-occur (Egger et al., 2016; Sela-Adler et al., 2017). Sulfate reduction is driven by the availability of hydrogen ($H_2$) (Muyzer and Stams, 2008) and is often coupled to anaerobic $CH_4$ oxidation (Valentine, 2002; Hoehler et al., 1994). Anaerobic $CH_4$ oxidation occurs in anoxic environments, such as coastal wetland or freshwater lake sediments, and can use various alternative electron acceptors, such as sulfate, nitrate, nitrite, humic substances, or metal oxides, for the oxidation of $CH_4$ (Haroon et al., 2013; Dang et al., 2021; Bai et al., 2019; Scheller et al., 2016; Valentine and Reeburgh, 2000). Aerobic $CH_4$ oxidation occurs mainly near the sediment surface or in the water column at the oxic-anoxic interface (King, 1992), where methanotrophs oxidize $CH_4$ to $CO_2$. In wetlands, sediments are often flooded or water-saturated, creating anoxic environments where oxygen ($O_2$) is limited or absent (Schlesinger and Bernhardt, 2013). Vegetated wetlands, however, such as those with reed plants, exhibit rhizospheric $CH_4$ oxidation influenced by the gas transport of the wetland plant (Brix et al., 2001; Armstrong and Armstrong, 1990; van der Nat and Middelburg, 1998). Taken together, these factors make examining wetlands with reeds (and sulfate) of particular interest.

Wetlands dominated by reed (*Phragmites australis*) can be a highly dynamic mosaic of reed, water, and sediment patches that vary due to seasonal or environmental factors (Buchsteiner et al., 2023; Baur et al., 2024b). Consequently, these ecosystems exhibit different interfaces and emission pathways for $CH_4$ fluxes with the atmosphere, not all of which are present in every season (Baur et al., 2024b). The plant-mediated transport of $CH_4$ via *P. australis* predominately occurs as pressurized convective gas flow (Armstrong and Armstrong, 1991; Brix et al., 2001), while molecular diffusion at the water–air and sediment–air interfaces represents another emission pathway. Additionally, ebullition as direct release of $CH_4$ bubbles from sediments, is another notable emission pathway. To date, only a few field studies have been conducted on all assessable emission pathways of wetlands with reed. An experiment was conducted in a German fen lake with high water levels on three days in June 2013, which suggested that plant-mediated transport plays a more significant role as ebullition (van den Berg et al., 2020). Another study, conducted in a subtropical Australian wetland partially covered with *P. australis*, demonstrated that the plant-mediated transport had the highest $CH_4$ emissions in only one of the two field campaigns (Jeffrey et al., 2019). The few available studies on this subject and the inconclusive results illustrate the necessity for further field studies on reed wetlands. Such research should incorporate measurements of all assessable emission pathways of each seasons. This would facilitate the determination of the precise contribution of plant-mediated $CH_4$ transport in comparison to the other emission pathways. A more comprehensive understanding of the various emission pathways of wetlands is essential to reduce uncertainty and account for the high variability associated with the processes of $CH_4$ production, transformation, and consumption in various types of wetlands before it reaches the atmosphere (Rosentreter et al., 2021). This understanding is particularly crucial for vegetated wetlands, where plant-mediated $CH_4$ transport plays a significant role (Vroom et al., 2022).

The $CH_4$ emissions in process-based vegetation models, such as LPJ-GUESS in Kallingal et al. (2024) or CLM4Me model in Riley et al. (2011), only partially implement the plant-mediated transport by considering only the passive mechanism (concentration gradient). Furthermore, these models do not account for pressurized flow, which occurs in plant species such as *Phragmites*, *Typha* sp., and others. However, it is known that $CH_4$ emissions from plant-mediated transport by *Phragmites australis* can be more than five times higher than by diffusion (Brix et al., 2001). According to Vroom et al. (2022), most other wetland greenhouse gas balance models do not consider plant-mediated $CH_4$ fluxes and exclude them in the total flux due to high variability in their contribution and the lack of data about this variability. This underscores the necessity of field studies in reed wetlands. Such studies could improve the parametrization and the implementation of emission pathways, such as the plant-mediated transport and, in particular, the pressurized flow mechanism, in the models. This could help to accurately model wetlands' greenhouse gas balances and reduce the global wetland $CH_4$ emission uncertainties (Vroom et al., 2022).

The diel (24 h) pattern of $CH_4$ emissions from aquatic macrophytes such as *P. australis* is highly influenced by the mode of gas transport and emission pathway (Chanton et al., 2002) and depends on the growth stage of the plant (Kim et al., 1998). In wetlands with reed, the diel variations in $CH_4$ emissions exhibited a wide range (2- to 5-fold) during the summer months (June–August/September), with the lowest emission occurring at night (Kim et al., 1998; van den Berg et al., 2016; Sanders-DeMott et al., 2022). However, the studies revealed discrepancies in the timing of the diel peak of $CH_4$ emissions during summer. Some studies have observed that this peak occurs around 12 o'clock (Kim et al., 1998; Sanders-DeMott et al., 2022; Zhang et al., 2016; van den Berg et al., 2016; Jeffrey et al., 2019), while others have reported that it occurs in the afternoon

(Baur et al., 2024b; Philipp et al., 2017). Moreover, in a wetland with reed that is subject to drought, the diel variation of the total ecosystem $CH_4$ emissions is less pronounced and exhibits a different diel pattern with two peaks, which is likely due to $CH_4$ oxidation (Baur et al., 2024b). However, these studies have often focused on the total $CH_4$ emissions of the ecosystem,

regardless of the emission pathways (van den Berg et al., 2016), or only on the summer season (van den Berg et al., 2020; Kim et al., 1998). However, there is a paucity of research examining diel patterns of each assessable emission pathway during all seasons, particularly in subsaline reed wetlands and in winter.

An example of a reed-rich wetland environment that is also subject to drought (Baur et al., 2024b), is Lake Neusiedl in Austria/Hungary. The reed ecosystem, which covers 181 km$^2$ (Csaplovics, 2019), occupies more than half of the lake and is

100 the largest contiguous reed area in Europe after the Danube Delta. The lake is subsaline (0.5–3 ‰, Hammer (1986)), rich in sulfate (250–1250 mg $SO_4^{2-}$ L$^{-1}$ in 2021, Baur et al. (2024a)), has no natural outflow, and is very shallow (< 1.8 m, Wolfram and Herzig (2013)). Lake Neusiedl is an international important wetland that is recognized by its cross-border protected areas, including a UNESCO World Heritage site, a Ramsar site, and the national park 'Neusiedler See – Seewinkel & Fertő – Hanság'. The presence of the main salt, sodium bicarbonate (Wolfram and Herzig, 2013), and sulfate creates specific environmental

conditions for reed ecosystems in inland waters (Baur et al., 2024a). Subsaline wetlands, especially subsaline reed ecosystems, are rarely ever studied. Due to their salinity, these ecosystems are more comparable to brackish ecosystems than to freshwater ecosystems. However, Lake Neusiedl's salt composition differs from these ecosystems, because another main salt is present besides sodium chloride. Overall, this underscores the need for microbial investigations and makes this wetland type interesting for studying carbon-related processes in different wetland types and conditions.

The objective of this study is to address the following research questions in the subsaline reed wetland of Lake Neusiedl: **(a)** Does the $CH_4$ flux show a distinct diel cycle in different emission pathways or seasons? **(b)** Which emission pathway dominates the diel $CH_4$ fluxes in which season, and what is its contribution? **(c)** To what extent can the microbial community or $\delta^{13}C$ source signatures explain the $CH_4$ formation or emission dynamics? In order to answer these questions, we carried out various chamber measurements during all seasons, determined isotopic source signatures, analyzed sediment properties, and

investigated the microbial community in the sediments.

## 2 Material and Methods

### 2.1 Study site

Lake Neusiedl (Fertő) on the Austrian–Hungarian border with an area of 315 km$^2$ (Wolfram and Herzig, 2013) is the western-most steppe lake in Europe with no natural outflow, but with an artificial outlet, which is closed since mid-2015 due to low

water levels (Baur et al., 2024b). The lake is very shallow, with declining annual maximum water levels from 1.66 m (2018) to 1.33 m (2022) due to drought (Baur et al., 2024b). The major salt of the subsaline lake is sodium bicarbonate (Wolfram and Herzig, 2013). More than half of the lake is covered by a reed ecosystem dominated by *Phragmites australis*. Due to drought, the mosaic of the reed belt has exhibited a notable increase in reed and open sediment areas, but also a corresponding loss of water areas during the years 2021 and 2022 (Buchsteiner et al., 2023; Baur et al., 2024b).

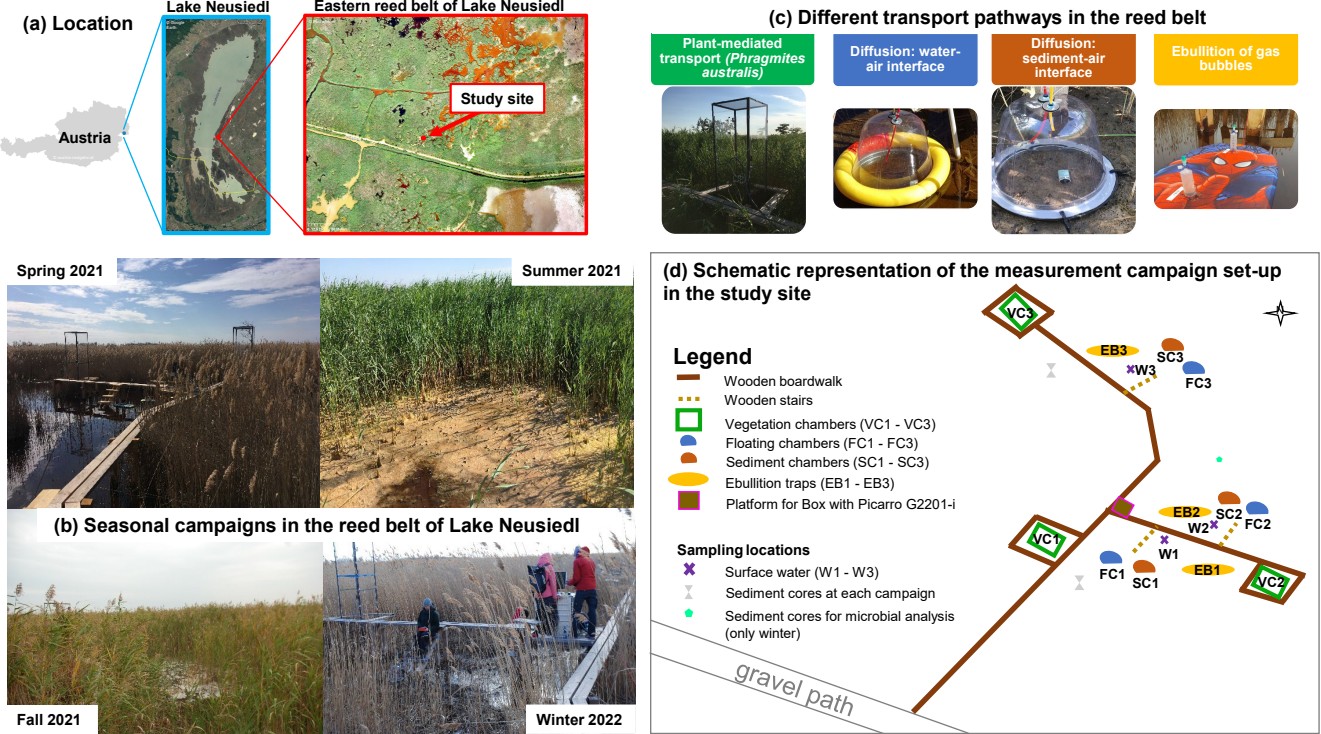

**Figure 1.** The study site is **(a)** located in the eastern reed belt of Lake Neusiedl in Austria and was used for **(b)** intensive 24 h measurement campaigns in each season. The different emission pathways of the CH$_4$ fluxes in the reed belt were measured with four chamber types illustrated in **(c)**. The setup of the measurement campaigns is shown in a schematic representation in **(d)**.

The study site (Lat: 47.7693°, Long: 16.7576°) is situated in the eastern reed belt of Lake Neusiedl near the Biological Station Lake Neusiedl (Illmitz, Austria) and in the nature zone of the national park (see Fig. 1a). In order to prevent any disturbance during the measurements, a boardwalk was constructed in the reed belt in December 2020. The site has a mean annual air temperature of 11.3°C and an annual precipitation of 500.2 mm (1993–2022) (Baur et al., 2024b).

### 2.2 Measurement setup

We conducted intensive 24 h measurement campaigns in each season (spring, summer, fall, winter; see Fig. 1b) to investigate the diel variability of CH$_4$ fluxes for each assessable emission pathway and their seasonal variability in the reed belt. As the reed belt is a dynamic mosaic of reed, water, and open sediment areas that vary according to season and condition, the reed belt has different exchange interfaces with the atmosphere, which are not always all available in every season (Baur et al., 2024b). For that reason, chamber and ebullition trap measurements were conducted for each emission pathway with the objective of investigating their respective contributions and dominance within the reed belt: plant-mediated transport of *P. australis*, diffusion at the water–air and sediment–air interfaces, and ebullition (see Fig. 1c). The spring campaign took place on 29 and

**Table 1.** The mean $\pm$ standard deviation of meteorological, water, and reed properties of the reed belt at our study site during the seasonal 24 h measurement campaigns ($T_{air}$ = air temperature, rH = relative humidity, LAI = leaf area index, $T_{water}$ = water temperature, WL = water level, CON = electrical conductivity, DO = dissolved oxygen, ORP = oxygen–reduction potential, Chl-a = Chlorophyll-a, $SO_4^{2-}$ = sulfate, and TOC = total organic carbon concentration of the surface water, if water was present). For LAI and WL, pure spatial mean values per campaign were used. For the other water parameters, three spatial replicates were used (location see Fig. 1d), which were sampled several times per campaign. For $T_{air}$ and rH, pure temporal mean values of the 24 h campaigns were used. The number of data points per parameter is given in parentheses.

| Season | DOY | Year | $T_{air}$ [°C] | rH [%] | LAI | $T_{water}$ [°C] | WL [cm] | $pH_{water}$ | $CON_{water}$ [mS cm⁻¹] | $DO_{water}$ [%] | $ORP_{water}$ [mV] | $Chl\text{-}a_{water}$ [µg L⁻¹] | $Sulfate_{water}$ [mg L⁻¹] | $TOC_{water}$ [mg L⁻¹] |
|---|---|---|---|---|---|---|---|---|---|---|---|---|---|---|
| spring | 88 & 89 | 2021 | 14.3 ± 3.1 (8640) | 57.8 ± 19.0 (8640) | 0.7 ± 0.4 (7) | 13.8 ± 1.6 (12) | 20 ± 3 (9) | 8.40 ± 0.04 (12) | 3.26 ± 0.02 (12) | 96.8 ± 9.6 (9) | 138.2 ± 45.4 (12) | 1.4 ± 3.1 (12) | 576.7 ± 12.6 (12) | 48.6 ± 2.8 (12) |
| summer | 179 & 180 | 2021 | 26.8 ± 6.5 (8640) | 64.1 ± 19.8 (8640) | 2.5 ± 1.1 (6) | 26.4 ± 5.9 (12) | 4 ± 3 (6) | 8.98 ± 0.04 (12) | 6.96 ± 0.42 (12) | 46.7 ± 68.7 (12) | −184.8 ± 115.2 (12) | 36 ± 33 (12) | 1366.1 ± 103.7 (12) | 132.1 ± 16.3 (12) |
| fall | 271 & 272 | 2021 | 15.9 ± 4.7 (8640) | 86.3 ± 12.5 (8640) | 1.4 ± 0.6 (8) | - | 0 | - | - | - | - | - | - | - |
| winter | 53 & 54 | 2022 | 5.4 ± 2.8 (8640) | 71.9 ± 17.3 (8640) | - | 6.1 ± 3.2 (12) | 4 ± 1 (6) | 8.91 ± 0.08 (12) | 6.02 ± 0.60 (12) | 97.2 ± 21.2 (12) | −34.2 ± 105.1 (12) | 36 ± 47 (12) | 1021.3 ± 160.1 (12) | 120.9 ± 14.8 (12) |

30 March 2021, the summer campaign on 28 and 29 June 2021, the fall campaign on 28 and 29 September 2021, and the winter campaign on 22 and 23 February 2022. To avoid a subjective selection of measurement days/nights, there was exactly a three-month period between the individual campaigns. The winter campaign was the only one that had to be postponed due to channel sediment excavation work in the study area, including local sediment deposits, which would have affected the measurements too much. In addition to seasonal water level fluctuations, there was a sharp decline in water levels in the study area, particularly in 2021 (and 2022; see Baur et al. (2024b)), indicating that the reed belt had experienced a prolonged dry period following the summer campaign. The meteorological conditions, water, and reed properties of the studied reed belt during the campaigns are pictured in Fig. 1b and summarized in Table 1. Table 1 summarizes the characteristics of each 24 h campaign and allows for comparison with other days/nights in the same season. During the study period, the surface water of the reed belt showed water levels ranging from 0 to 23 cm (see Table 1). If water was present, pH values ranged from 8.4 to 9 and electrical conductivity from 3.3 to 7 mS cm$^{-1}$.

The measurement intervals in a 24 h measurement campaign were every 3 h for each chamber type and every 6 h for gas bubbles in the ebullition traps. There were 3 replicates of each chamber type and 9 replicates of ebullition traps (funnels) spatially distributed around the constructed boardwalk at the study site (see Fig. 1d). In situ measurements of surface water were conducted, and samples were collected every 6 h in each season when water was available above the surface (see Section 2.3 for details). Sediment cores were sampled in each campaign and analyzed in the laboratory for chemical and physical properties (see Section 2.4.1 for details). In order to determine the microbial status in the sediments and its changes over time, two sediment cores were taken before the first and after the last measurement campaign (one year apart) and are described in Section 2.4.2.

### 2.2.1 Pathways divided by different chamber measurements and ebullition traps

Different measurement types were used to capture the different emission pathways of $CH_4$ fluxes in the reed belt individually: plant-mediated transport of *P. australis* with vegetation chambers (VC), diffusion at the water–air interface with floating

chambers (FC), diffusion at the sediment–air interface with sediment chambers (SC), and ebullition with ebullition traps (EB) (see Fig. 1c). The setup of the campaigns with the location of the chambers and the sampling points is schematically represented in Fig. 1d.

VCs were used to capture the plant-mediated transport of *P. australis*. For each VC, a steel frame (2.99 m × 0.78 m × 0.78 m) was welded with a ground frame (0.2 m × 0.78 m × 0.78 m), which was anchored permanently at the study site since early March 2021 and inserted at least 7 cm into the ground. At the beginning of each campaign, every VC was covered with transparent acryl glass plates (3 walls and 1 lid), which were all surrounded with highly strong magnetic stripes. A mobile acryl glass plate equipped with an inlet tube plug-in connector, an outlet tube plug-in connector, and a power cable was used to close the chamber during measurements. Each VC had three fans at three different heights to ensure good ventilation inside the chamber.

FCs were used to capture the transport of molecular diffusion at the water–air interface and are described in detail in Baur et al. (2024a). SCs were used to capture the molecular diffusion transport at the sediment–air interface. The SC top was made of transparent PVC in the form of a growing bell with a diameter of 30 cm and a hard plate with sealing rubber on the base. The bottom frame was an aluminum ring surrounded by a sealing rubber, which was inserted to the sediments 2–3 weeks in advance of each campaign. The upper chamber and the bottom frame were fixed and held together with at least 5 clamps during each measurement.

In VC, FC, and SC, the dry mole fraction of $CH_4$ (sum of $^{13}CH_4$ and $^{12}CH_4$) and the stable carbon isotope ratios ($\delta^{13}C$-$CH_4$ and $\delta^{13}C$-$CO_2$) of the chamber air were measured directly in the field during chamber closure with an isotopic gas analyzer (G2201-i, Picarro Inc., Santa Clara, USA) using cavity ring-down spectroscopy (CRDS; highly sensitive laser absorption technology) and with an external diaphragm vacuum pump (MD 1, Vaccuubrand GmbH, Wertheim, Germany) for the CRDS. During the field measurements, the simultaneous measurement mode for $CO_2$ and $CH_4$ as well as the high precision mode of the Picarro instrument were used, which has a measurement interval of about every 5 sec. During measurements, SC, VC, and FC were directly connected to the Picarro instrument through two 6 mm outer diameter polyurethane tubes (Festo GmbH, Esslingen, Germany), one as inlet and another as outlet, which circulated the air between the closed chamber and the instrument. For air circulation, an additional KNF diaphragm gas pump (NMP830KNDC, KNF Micro AG, Reiden, Swiss) and an external air flow controller (2510A2A13BVBN, Brooks Instrument, LLC, Hatfield, USA) with a constant flow rate of 0.9 LPM were used. The closing time for FC and SC was 5 min each, the VC closing time was 10 min due to the larger chamber volume. All isotopic ratios measured in this study are expressed in ‰ relative to Vienna Pee Dee Belemnite (VPDB). Before each campaign, the instrument was calibrated using two certified stable isotope standard gases, which are described in Baur et al. (2024a). The standard deviation of $\delta^{13}C$-$CH_4$ during the field measurements (uncertainty) was, on average, 3.5 ‰ VPDB, while the standard deviation of $\delta^{13}C$-$CO_2$ was 0.6 ‰ VPDB.

EBs were used to capture the ebullition pathway of gas bubbles from supersaturated sediments and are described in detail in Baur et al. (2024a). This study of Baur et al. (2024a) examined the ebullition pathway continuously over a more extended period (from March to July 2021) at the same lake and not solely within the reed belt. The $\delta^{13}C$-$CH_4$ values and the dry mole fraction of $CH_4$ of the collected gas from EB were measured with the same Picarro instrument, but with an additional small

sample isotope module (SSIM2) (A0314, Picarro, Inc., Santa Clara, CA, USA) including an additional external diaphragm
vacuum pump (MD 1, Vaccuubrand GmbH, Wertheim, Germany) for the SSIM in the lab shortly after each campaign (for
further details, please refer to Baur et al. (2024a)). The SSIM enables syringe injection of small gas sample volumes with
measurement repetitions. With the SSIM, the fast method was used with two measurement repetition per gas sample and a pure
measurement duration of 8 min for recording an average value per gas sample repetition. The standard deviation of $\delta^{13}$C-CH$_4$
during the measurements (uncertainty) was, on average, 0.25 ‰ VPDB. Due to the high CH$_4$ concentration of ebullition gas
bubbles, the high dynamic range mode of the Picarro had to be applied in the lab.

During the spring campaign, there were no open sediment areas at the study site due to higher water levels within the
reed belt, which precluded the possibility of performing SC measurements. During the course of the year, due to the sharp
drop in water levels not only in the open lake area of Lake Neusiedl, but also in the reed belt (Baur, 2024), EB and FC
measurements were not possible in fall, as there was no water level above the surface. Furthermore, it was not feasible to
conduct EB measurements during the winter campaign, as the exceedingly low water levels (and small water areas) precluded
the installation of the inverted funnels, which would have entailed contact with and disturbance of the sediments. The EB
measurements in summer were only possible despite the low water level and before the site ran dry, because the traps had
already installed at the beginning of March 2021, when the water level was high.

### 2.2.2 Flux calculation

For the calculation of the fluxes of the chamber measurements (FC, SC, and VC), the initial 25 % of the chamber closure time
(death band) was removed from the linear regression fit of the temporal change of the dry mole fractions of CH$_4$ to exclude
potential artifacts due to chamber closing (Hoffmann et al., 2017). Ebullition (a sudden exponential increase in the dry mole
fraction) during chamber measurements was excluded for the calculation of diffusive fluxes. Due to the presence of a highly
agile sediment layer (soft and waterlogged), ebullition was sometimes triggered by the closure of the SC during the summer
campaign. However, since the aim was to measure diffusion rates at the sediment–air interface with the SC, a substantial
proportion of the SC measurement data from summer had to be excluded. The equations utilized to estimate chamber fluxes
and ebullition rates are thoroughly delineated in Baur et al. (2024a). Only flux data from FC, SC, and VC were utilized that
exhibited a coefficient of determination $R^2$ of at least 0.6 for the linear regression of the temporal change of the dry mole
fractions of CH$_4$, or in instances where the CH$_4$ flux was minimal (near zero), i.e. between $-0.01$ and 0.01 mg CH$_4$ m$^{-2}$ h$^{-1}$ and
the $R^2$ was at least 0.6 for the temporal change of the dry mole fraction of CO$_2$. In total, 213 chamber CH$_4$ fluxes were used,
with a mean $R^2$ of 0.9. Additionally, 72 ebullition rates were calculated across two season campaigns (spring and summer), of
which 26 exhibited ebullition rates exceeding zero.

### 2.2.3 Stable carbon isotope ratios

The Keeling plot approach was applied to determine the seasonal source signatures of $\delta^{13}$C (Keeling, 1958; Pataki et al., 2003)
because transport and oxidation fractionate the stable carbon isotopes (Conrad, 2005; Chanton, 2005). The source signature of
CH$_4$ is the intercept of the linear regression between the reciprocal of the dry mole CH$_4$ fraction and the values of $\delta^{13}$C-CH$_4$.

The values used in the Keeling plot are the mean values of the chamber air (of VC, FC, or SC) during the chamber closure of each individual chamber measurement (5–10 min depending on the chamber type, see Section 2.2.1). The $CO_2$ source signature was determined in a similar manner, albeit with $CO_2$ values. Subsequently, with the $\delta^{13}C$ source signatures of $CH_4$ and $CO_2$ of each season, the methanogenic characterization according to Whiticar et al. (1986) was applied to determine the dominant methanogenic pathway of each season and the contribution of $CH_4$ oxidation. Since ebullition is known not to fractionate the $\delta^{13}C$-$CH_4$ values and to bypass the $CH_4$ oxidation region (Chanton, 2005), the measured values of the collected ebullition gas can directly show the isotopic source signature of $CH_4$ within the sediments without applying the Keeling plot approach.

### 2.2.4 Environmental variables

Each SC and FC was equipped inside with an air temperature and a light sensor (HOBO Pendant Temperature/Light 8K Data Logger, Onset Computer Corporation, Bourne, USA) with a measurement time interval of 30 sec. Each VC was equipped inside with an air temperature and relative humidity sensor (HOBO U23 Pro v2 Temperature/Relative Humidity Data Logger, Onset Computer Corporation, Bourne, USA) with a measurement time interval of 10 sec. Under each floating board with three ebullition traps, a water temperature sensor (HOBO Pendant Temperature/Light 8K Data Logger, Onset Computer Corporation, Bourne, USA) was fixed with a measurement time interval of 30 min due to the smaller data memory. Additionally, ambient air temperature, water vapor pressure deficit (VPD), incoming shortwave radiation ($SW_{in}$), and sediment temperature (in 5 cm depth) were measured nearby (50 m distance) at an eddy covariance tower (detailed description in Baur et al. (2024b) and published data in Baur (2024)).

### 2.3 Water sampling and analysis

During the 24 h campaigns, the following parameters were measured in situ in three surface water areas (sampling location Fig. 1d) every 6 h using a portable WTW multiparameter probe (Multi 3420, Xylem Analytics GmbH, Weilheim, Germany): pH, electrical conductivity (CON), water temperature ($T_{water}$), dissolved oxygen (DO), and oxygen–reduction potential (ORP). At the same time interval, three surface water samples were collected, stored in a cool box, and analyzed immediately after the campaign in the accredited laboratory of the nearby Biological Station Lake Neusiedl (Illmitz, Austria). In the laboratory, sulfate concentrations were analyzed according to DIN EN ISO 10304-1 (DIN, 2009) using ion chromatography (Dionex ICS 1100, Thermo Fisher Scientific Inc., Waltham, USA). The chlorophyll-a concentration (Chl-a) was determined by the LORENZEN method (Goltermann, 1969). Total organic carbon concentration (TOC) was determined according to DIN EN 1484 (DIN, 2019) by catalytic combustion oxidation using the TOC-L analyzer (Shimadzu Corp., Kyoto, Japan).

### 2.4 Sediment sampling and analysis

In each campaign, at least two sediment cores were sampled with polycarbonate push core tubes (diameter 7 cm, length 60 cm). Each sediment core was divided according to its two visibly distinct layers for physical and chemical analysis. Fresh samples of each layer were sieved with a 2 mm sieve to remove large rhizomes, large roots or snail shells. Additionally, two sediment

cores for microbial analysis were sampled in the reed belt always in winter before the first (11 February 2021) and after the last campaign (23 February 2022) using push core tubes, which were sealed with Parafilm after sampling. The sediment cores were divided into 2 cm depth sections in an anaerobic tent filled with nitrogen ($N_2$), individually packed, sealed, and stored at 4°C in a refrigerator until laboratory analysis. The microbial analysis was carried out in four specific depth sections of each sediment core, two of which belong to the same layer: 0–2 cm, 4–6 cm, 20–22 cm, 24–26 cm.

### 2.4.1 Physical and chemical sediment characterization

The gravimetric water content (WC) of the sediments was determined by completely drying the accurately weighed field wet sample in a freeze dryer (Lio 5P, Kambic d.o.o., Semic, Slovenia) until constant dry weight was reached and calculated according to Gardener (1986). The completely dry sediment samples were finely ground (pulverized) and homogenized using a mixer mill (MM400, RETSCH GmbH, Haan, Germany). A multiphase carbon analyzer (RC612, LECO Europe B.V., AG Geleen, Netherlands) was used to determine the content of organic (TOC, $\leq 450°C$), inorganic (TIC, $> 450°C$ & $\leq 1000°C$), and total carbon (TC, sum of TOC and TIC) of the finely ground samples (in percentage of dry mass) via temperature-dependent $CO_2$ measurement (dry combustion method).

The analysis of sulfate, ammonium, nitrate, nitrite, and orthophosphate in the sediments was conducted by preparing sediment extracts from 4 g field wet sediment samples and 30 mL extraction solution (0.01 M $CaCl_2·2H_2O$ for sulfate; ultrapure distilled water for the other parameters). The sediments were extracted on a horizontal shaker at 300 rpm for 60 min at room temperature to obtain optimal suspension of the slurry. The extracts were filtered through 0.45 $\mu$m nylon syringe filters. The concentrations of the extracts were determined by turbidity or colorimetry using a UV-vis spectrometer (Infinite 200 Pro NanoQuant, Tecan Group Ltd., Männedorf, Switzerland). The specific methods are described in detail in Baur et al. (2024a).

For the pH and CON, the field wet sediment samples were air dried at 40°C until mass stability was achieved. The samples were then incubated overnight with ultrapure distilled water (ratio 1:10 w/v), shaken in an overhead shaker for 1 h, and allowed to stand for 30 min before pH (pH meters 7110 and 720) and CON (Cond 7110) were measured (Xylem Analytics GmbH, Weilheim, Germany).

### 2.4.2 Microbial 16S rRNA gene amplification, sequencing and analysis

Each sediment depth section sample, weighing between 0.5 to 1.0 g, underwent DNA (deoxyribonucleic acid) extraction utilizing the DNeasy PowerSoil Pro Kit (Qiagen NV, Venlo, Netherlands), following the manufacturer's guidelines. Subsequently, DNA concentration was determined using a Qubit 2.0 Fluorometer (Invitrogen, Carlsbad, USA) equipped with the dsDNA HS kit. For amplification of prokaryotic 16S rRNA genes, we employed PCR (polymerase chain reaction) with the primer pair 515f (5'- GTG CCA GCM GCC GCG GTA A) and 806r (5'- GGA CTA CHV GGG TWT CTA AT), as detailed previously (Caporaso et al., 2011). The PCR products were then barcoded and sequenced at the Vienna BioCenter Core Facilities (VBCF) on the Illumina Miseq platform (300 PE). The initial raw reads underwent preprocessing using cutadapt (Martin, 2011) to eliminate primer sequences, followed by sequence analysis via the QIIME2 pipeline (Bolyen et al., 2019). In summary, the data was denoised and filtered for low-quality reads and chimeras using the DADA2 algorithm. Subsequently, sequences exhibiting

100 % identity were clustered into amplicon sequence variants (ASVs). Taxonomic classification of these ASVs was conducted utilizing the SILVA database (release 138) in conjunction with the 'q2-feature-classifier' plugin (Bokulich et al., 2018).

## 2.5 Data and statistical analysis

Most data and statistical analysis were performed in R, version 4.2.2 (R Core Team, 2022), using the following R packages for data processing, manipulation and visualization: *data.table, ggplot2, ggpubr, scales, hms* (Müller, 2023; Barrett et al., 2024; Kassambara, 2023a; Wickham, 2016; Wickham et al., 2023). The two-sided Dunn's post-hoc test was applied using the R package *rstatix* (Kassambara, 2023b) to statistically assess which emission pathway or season had significant differences in median $CH_4$ fluxes compared to the others and in sediment properties between the two sediment layers ($p < 0.05$, Adjustment Holm). Linear ordinary least squares regression using the R package *lmodel2* (Legendre, 2018) was used to determine the $\delta^{13}C$ source signatures in Keeling plots.

Potential interactions between the methanogenic archaea and other microbes were investigated by employing SparCC (Sparse Correlations for Composition data) network (Friedman and Alm, 2012) analysis at the ASVs level. Only those with > 10 reads and detected in > 20 % of the samples were used for network construction to prevent false associations caused by low abundant ASVs. 1000 bootstraps were used and those with $p < 0.05$ were retained. The positive relationships (correlation coefficient > 0) were summarized at the class level. Only the top 10 groups with highest number of potential interactions with methanogenic archaea were shown, with network visualization being conducted in Cytoscape (Shannon et al., 2003).

## 3 Results

### 3.1 Seasonal and diel differences in $CH_4$ emissions of a subsaline reed wetland

Plant-mediated transport exhibited significantly elevated $CH_4$ emissions compared to emissions through diffusion or ebullition pathways throughout all seasons, as illustrated in Table 2. Regardless of the emission pathway, the highest median $CH_4$ emission rate and fluctuation range occurred during the summer campaign. The highest $CH_4$ emission of 18.56 mg $CH_4$ $m^{-2}$ $h^{-1}$ was observed via plant-mediated transport at 15 o'clock during the summer campaign. In summer, the median $CH_4$ emissions at the two diffusion interfaces (water–air and sediment–air) and the ebullition pathway did not show significant differences. However, in winter, the water–air interface exhibited a median $CH_4$ flux of 0.05 mg $CH_4$ $m^{-2}$ $h^{-1}$, which was more than 10-fold and significantly higher than that observed at the sediment–air interface. Almost all emission pathways exhibited a significantly different median $CH_4$ flux between the seasons. Only at the sediment–air interface, the median flux did not differ significantly between fall and winter. The ebullition pathway exhibited the most pronounced fluctuations in $CH_4$ emissions across the two seasons in which EB measurements were possible, and demonstrated the lowest emission rate in spring. In spring, bubbles were trapped in only 3 % of the EB measurements, while in summer it happened in 69 %. The measurement of all pathways was not possible in all seasons due to variations in water levels and the dryness of the reed belt (see Fig. 1).

**Table 2.** Differences in CH$_4$ fluxes for each emission pathway (plant-mediated, water–air interface, sediment–air interface, ebullition) and season are represented by their median ± standard deviation; significant differences ($p < 0.05$ with Holm's adjustment, Dunn's test) between the pathways in each season are indicated by different superscript letters, between the seasons in each pathway by different superscript symbols or between daytime and nighttime in each pathway and season by different superscript numbers; the number of quality checked measurement data is provided in brackets; due to the water level or dryness of the reed belt, not every pathway could be measured in each season; nighttime when incoming shortwave radiation was < 10 W m$^{-2}$.

| | | CH$_4$ flux [mg CH$_4$ m$^{-2}$ h$^{-1}$] | | | |
| --- | --- | --- | --- | --- | --- |
| Time period | Pathways | Spring 2021 | Summer 2021 | Fall 2021 | Winter 2022 |
| diel (24 h) | plant-mediated | $1.36 \pm 0.23^{a,+}$(24) | $4.59 \pm 3.61^{d,*}$(24) | $0.35 \pm 0.30^{f,\otimes}$(24) | $0.12 \pm 0.06^{h,\wedge}$(24) |
| | water–air | $0.11 \pm 0.07^{b,\cap}$(23) | $1.12 \pm 0.51^{e,\pi}$(16) | - | $0.05 \pm 0.06^{i,\cup}$(23) |
| | sediment–air | - | $1.21 \pm 0.52^{e,\backslash}$(08) | $0.002 \pm 0.01^{g,\oplus}$(24) | $0.004 \pm 0.09^{k,\oplus}$(23) |
| | ebullition | $0.000 \pm 1.00^{c,\Omega}$(36) | $1.35 \pm 3.69^{e,\Phi}$(36) | - | - |
| daytime | plant-mediated | $1.54 \pm 0.23^{a,+,1}$(13) | $6.37 \pm 3.90^{d,*,2}$(15) | $0.52 \pm 0.38^{f,+\wedge,4}$(10) | $0.11 \pm 0.05^{h,\wedge,5}$(9) |
| | water–air | $0.14 \pm 0.08^{b,\cap,6}$(12) | $1.07 \pm 0.42^{e,\pi,8}$(12) | - | $0.06 \pm 0.02^{hk,\cup,9}$(9) |
| | sediment–air | - | $1.21 \pm 0.48^{e,\backslash,10}$(06) | $0.001 \pm 0.01^{g,\oplus,11}$(11) | $0.006 \pm 0.10^{k,\otimes,12}$(11) |
| | ebullition | $0.000 \pm 1.41^{c,\Omega,13}$(18) | $1.28 \pm 3.77^{e,\Phi,14}$(27) | - | - |
| nighttime | plant-mediated | $1.23 \pm 0.22^{a,+,1}$(11) | $3.19 \pm 1.60^{d,+,3}$(09) | $0.29 \pm 0.21^{f,\wedge,4}$(14) | $0.12 \pm 0.06^{h,\wedge,5}$(15) |
| | water–air | $0.09 \pm 0.04^{b,\pi\cup,7}$(11) | $1.59 \pm 0.60^{d,\pi,8}$(04) | - | $0.03 \pm 0.08^{hk,\cup,9}$(14) |
| | sediment–air | - | $1.48 \pm 0.84^{d,\oplus,10}$(02) | $0.002 \pm 0.01^{g,\oplus,11}$(13) | $0.001 \pm 0.08^{k,\oplus,12}$(12) |
| | ebullition | $0.000 \pm 0.00^{c,\Omega,13}$(18) | $2.31 \pm 3.59^{d,\Phi,14}$(09) | - | - |

Only plant-mediated transport showed a clear visible diel cycle of CH$_4$ fluxes in summer with the highest emission rates during daytime and on average a peak in the late afternoon (see Fig. 2). In contrast, the diffusion pathways showed only some small diel variations, but no distinct diel cycle. However, we were able to detect a decrease in diffusive CH$_4$ fluxes at the water–air interface around midday in summer. In spring and fall, only plant-mediated transport displayed some diel variations of CH$_4$ fluxes, but not the diffusion pathways. Significant differences in the median CH$_4$ flux between daytime and nighttime (SW$_{in}$ < 10 W m$^{-2}$) were identified for only two emission pathways, occurring in one season each: water–air in spring and plant-mediated in summer (see Table 2). Given that the ebullition trap measurements were consistently obtained over a 6 h period, we have not plotted a fitted line of diel variations, as the precise timing of bubble release from the sediments remains uncertain. Instead, the observed CH$_4$ ebullition rate was placed in the middle of the 6 h bubble collecting time in Fig. 2. During the spring campaign, only one ebullition rate of 5.98 mg CH$_4$ m$^{-2}$ h$^{-1}$ could be observed during daytime. In contrast, during the summer campaign, ebullition occurred at nighttime and during daytime, with the highest rate of 17.03 mg CH$_4$ m$^{-2}$ h$^{-1}$.

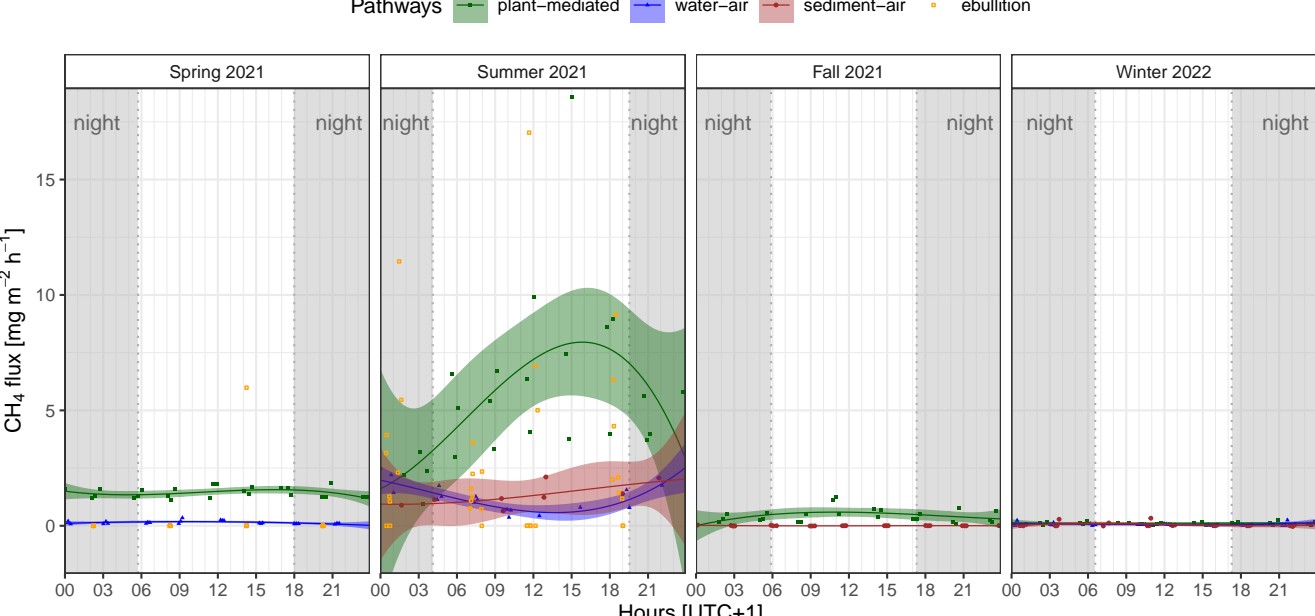

**Figure 2.** Diel variability of the CH$_4$ fluxes for each season and emission pathway (plant-mediated, water–air interface, sediment–air interface, ebullition); each point represents an individual measured flux rate; the fitted lines show the 3$^{rd}$ order polynomial regressions with the 95 % confidence interval; due to the water level or dryness of the reed belt, not every pathway was present or could be measured in each season; nighttime when incoming shortwave radiation was < 10 W m$^{-2}$ (gray shading).

## 3.2   Seasonal $\delta^{13}$C source signatures of a subsaline reed wetland

The Keeling plots' source signatures $\delta^{13}$C-CH$_4$ differ between seasons and are most $^{13}$C-depleted in fall ($-73.6 \pm 2.4$ ‰ VPDB; see Fig. 3a). The Keeling plots' source signatures $\delta^{13}$C-CO$_2$ differ between seasons and are most $^{13}$C-depleted in winter
($-30 \pm 1.6$ ‰ VPDB; see Fig. 3b). The methanogenic characterization after Whiticar et al. (1986) indicates the dominance of acetoclastic methanogenesis in the reed belt of Lake Neusiedl regardless of the seasons (see Fig. 3c). However, the values of the source signatures including the standard error of the winter campaign are already partially within the range of CH$_4$ oxidation in this representation. For this subsaline reed wetland, the mean of the seasonal isotopic source signatures are $-63.6 \pm 2.7$ ‰ VPDB for CH$_4$ and $-28.1 \pm 1.2$ ‰ VPDB for CO$_2$. The $\delta^{13}$C-CH$_4$ values of the collected ebullition gases were $-54.2$ ‰ VPDB ($N = 1$) in the spring campaign and $-59.9 \pm 1.9$ ‰ VPDB ($N = 25$) in the summer campaign. In fall
and winter, no ebullition trap measurements were possible due to the dryness of the reed belt.

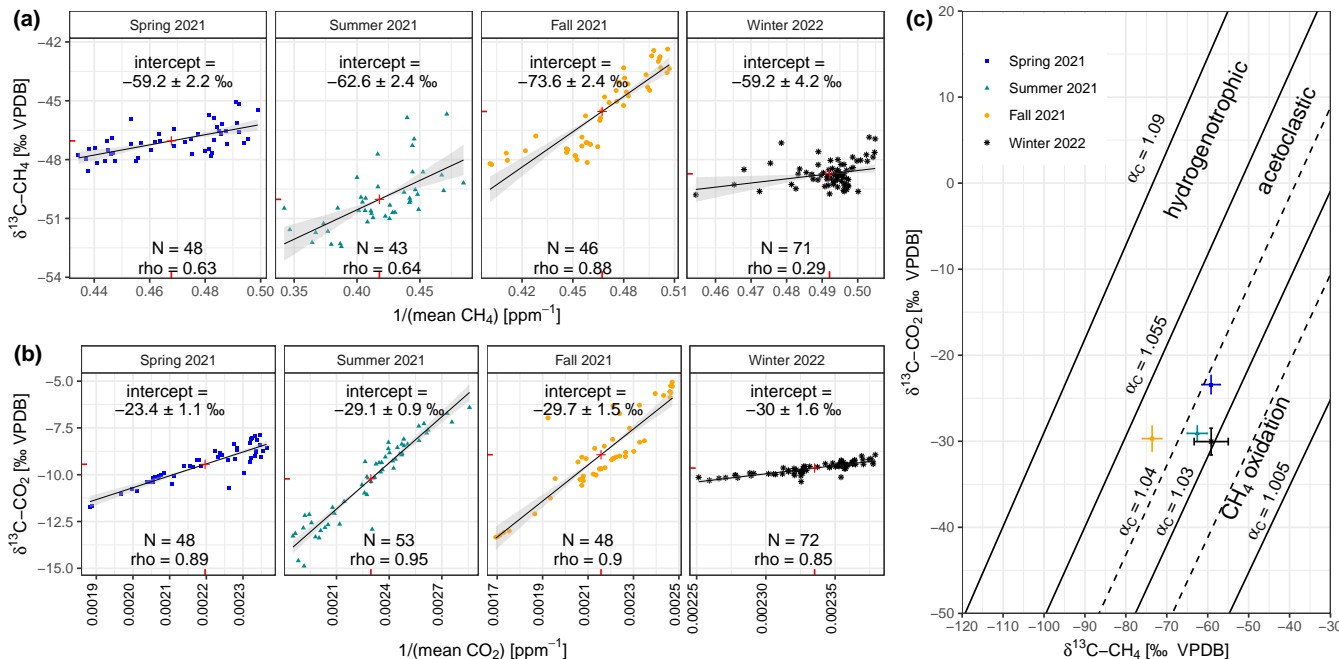

**Figure 3.** Keeling plots to determine the seasonal source signatures **(a)** $\delta^{13}$C-CH$_4$ and **(b)** $\delta^{13}$C-CO$_2$ of the reed belt as the y axis intercepts (shown as number in ‰ VPDB) of the fitted ordinary least square linear regression lines (black) with 95 % confidence interval (shaded light gray) and their centroids (red cross) using the mean chamber air values during the chamber closure, and **(c)** the methanogenic characterization according to Whiticar et al. (1986) using the calculated source signature pairs of $\delta^{13}$C-CH$_4$ and $\delta^{13}$C-CO$_2$ and their standard errors from the plots **(a)** and **(b)**.

## 3.3 Sediment properties of the reed belt

The sediments of the reed belt of Lake Neusiedl had two clear distinguishable layers with significant differences in most physical and chemical parameters (see Fig. 4 and A3). The upper layer (L1) was very wet (mean ± standard deviation of gravimetric WC: 82.3 ± 6.7 mass-%, see Fig. 4b), brown, peaty (mean TOC: 18.2 ± 9.6 mass-%, see Fig. 4a), and with a mean thickness of 9.7 ± 3.8 cm. The lower layer (L2) was gray, sandy, and low in organic carbon (mean TOC: 0.8 ± 0.6 mass-%). In spring, L1 was a fluffy layer, had the highest water level above it (20 ± 3 cm), and had occasionally a maximum thickness of 19 cm. However, there were no significant differences in L1 thickness between seasons. In contrast, the TOC and TIC values of L1 in particular differed between the seasons (see Fig. 4a and A3a). The mean TIC decreased from spring to winter, while the mean TOC increased during the same period, except for the mean value of the summer season. The SO$_4^{2-}$ concentration in the sediments showed the same seasonal pattern as the TOC values (see Fig. 4a and c). L1 showed the highest concentration of NH$_4^+$ in summer and the highest concentration of NO$_2^-$ in fall (see Fig. A3d and f). The upper sediment layer exhibited, on average, a significantly higher CON value and a significantly lower pH value compared to the lower layer (see Fig. A3g and h).

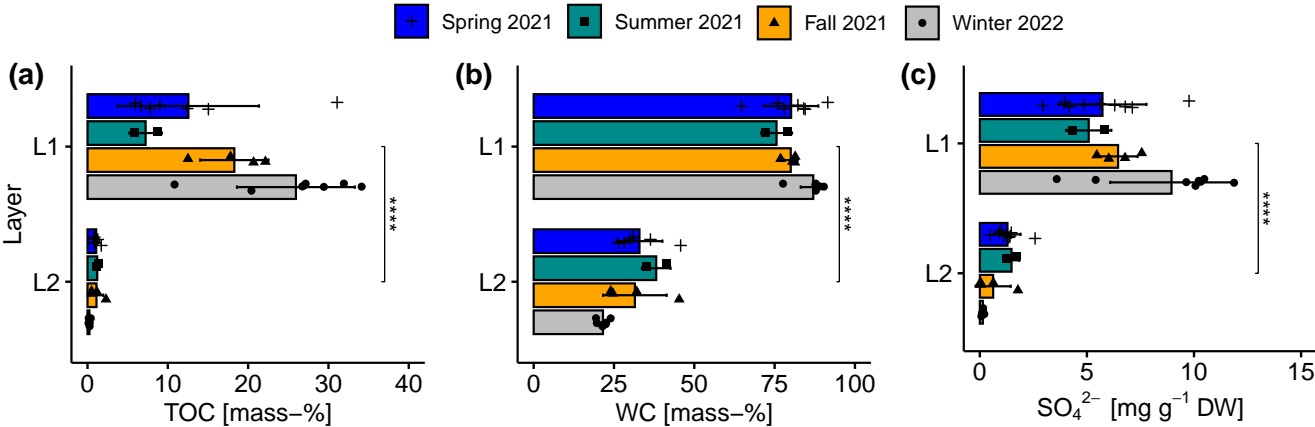

**Figure 4.** Seasonal and layer specific differences of **(a)** total organic carbon content (TOC), **(b)** gravimetric water content (WC), and **(c)** sulfate ($SO_4^{2-}$) concentration in the upper (L1) and lower layer (L2) of sediments in the reed belt of Lake Neusiedl. Very significant differences (****, $p < 0.0001$ with Holm's adjustment, Dunn's test) in the parameters between the two layers in the sediments independent of the season.

### 3.4 Change in the microbial community in the sediments within one year

Considering the findings regarding $CH_4$ fluxes throughout 2021, and acknowledging the central role the microorganisms play in $CH_4$ emission dynamics, our study investigated the microbial community composition within sediments from Lake Neusiedl at two different sampling dates: one in winter 2021 and another a year later in winter 2022. We conducted our analysis across various depths in two sediment cores each winter, employing prokaryotic 16S rRNA gene amplicon sequencing. It is worth pointing out that there was a marked decrease in total DNA yield in samples from the lower layers in comparison to those

from the upper layers, with a decrease of almost 1000-fold in 2021 and almost 32-fold in 2022 (see Table A1). Due to the significance of sulfate-reducing bacteria, methanogens, and methanotrophs in carbon cycling within anaerobic sediment environments (Jones, 1985; Conrad, 2007; Muyzer and Stams, 2008), we specifically targeted these three groups for further investigation (see Fig. 5). The amplicon sequencing quality of core II at the depth of 4–6 cm from the 2022 sampling was insufficient for its inclusion in further analyses.

Examination of the distribution pattern of methanogens (see Fig. 5a) reveals a noticeable trend: their relative abundance is higher in the lower depths (20–22 cm and 24–26 cm) samples compared to the upper layer in both years, although this difference is less pronounced in the 2021 sampling. When comparing their diversity across the two years, various observations emerge. First, in 2022, their relative abundance was substantially higher, reaching up to 12 % compared to a maximum of around 2 % in 2021. Second, the composition of methanogenic communities differed substantially. In 2021, sediments exhib-

ited a broader array of detected methanogenic groups, encompassing orders such as *Methanobacteriales*, *Methanosarcinales*, *Methanococcales*, *Methanomicrobiales*, and *Methanomassiliicoccales*. Conversely, in 2022, the methanogenic community displayed greater homogeneity, predominantly dominated by organisms from the order *Methanomicrobiales*.

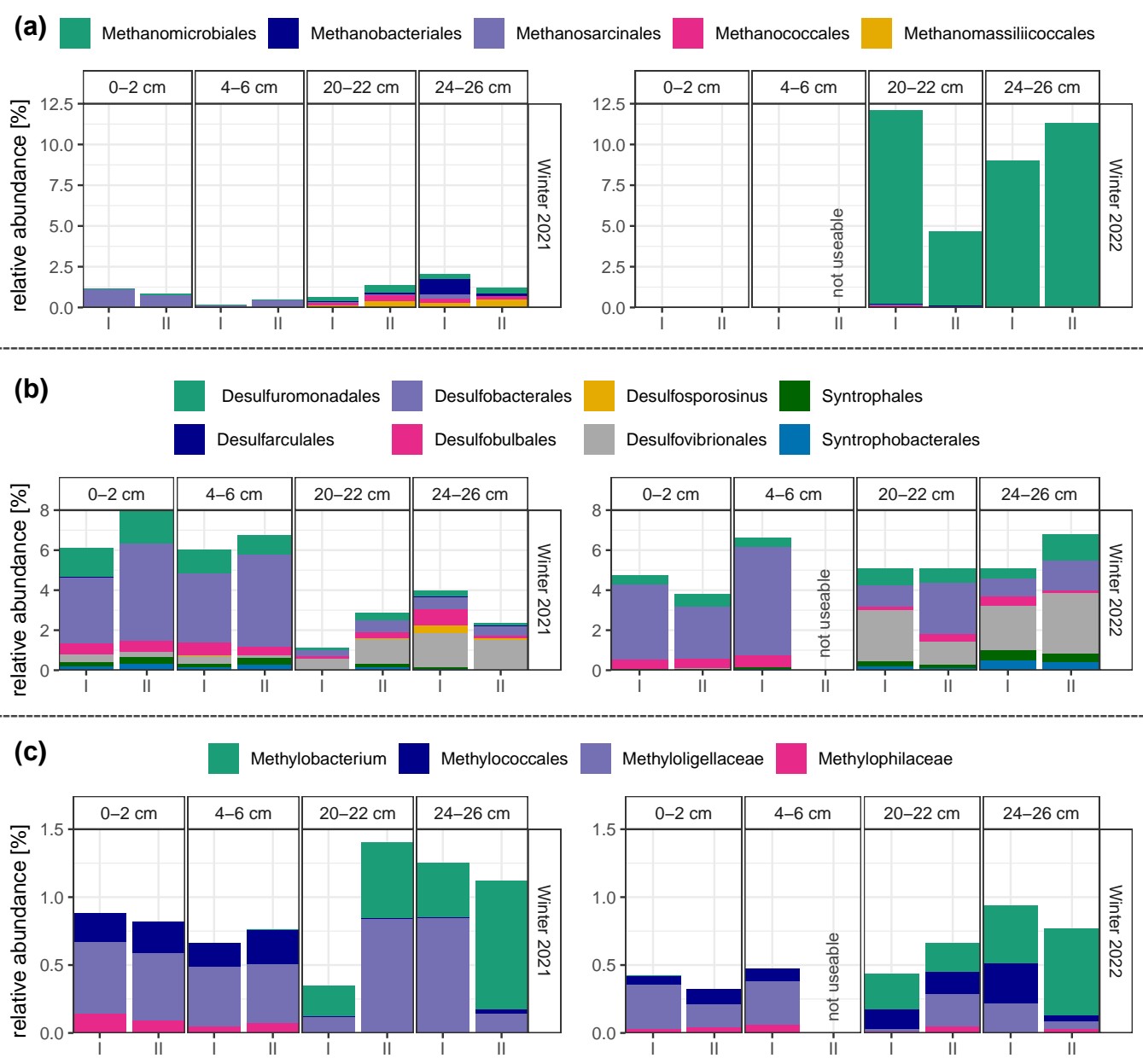

**Figure 5.** Relative abundance of **(a)** methanogens, **(b)** sulfate-reducing bacteria, and **(c)** methanotrophs in four depth sections of the two sediment cores (I and II) in winter 2021 and 2022. The two sediment sections, 0–2 cm and 4–6 cm, are part of the top organic layer (L1) of the sediment, while the sections 20–22 cm and 24–26 cm are located in the lower sandy layer (L2). Due to the insufficient amplicon sequencing quality of core II at the depth of 4–6 cm from the 2022 sampling, the sample was not used for further analyses.

The relative abundances of sulfate-reducing bacteria (see Fig. 5b) display a distinct trend in the year 2021. There is a marked discrepancy between the upper (0–2 cm and 4–6 cm) and the lower layer (20–22 cm and 24–26 cm). In the upper layer, the

relative abundance of sulfate-reducing bacteria is higher, reaching levels of up to around 8 %, in contrast to the deeper layer where it reaches numbers of around 4 %. This difference in abundance is accompanied by a visible diversity pattern. The upper layer predominantly hosts taxa such as *Desulfobacterales*, *Desulfuromonadales*, and *Desulfobulbales*. While these groups persist in the lower layer, there is a shift towards an increased prevalence of organisms belonging to the order *Desulfovibrionales*, eventually becoming the most abundant sulfate-reducing bacterial group in the 24–26 cm depth. The sediments sampled in

2022 present a different pattern, with comparable sums of relative abundances of sulfate reducers across different depths in 2022 in contrast to 2021. However, the dominant taxa are similar to those observed in the previous year, where the same groups are also prevalent in the upper layer, and *Desulfovibrionales* becomes more abundant as depth increases.

Analysis of methanotroph distribution reveals similar patterns across samplings from both 2021 and 2022 (see Fig. 5c). In both years, a clear distinction is observed between the upper and lower layer, with *Methylobacterium* being detected ex-

clusively in depths $\geq 20$ cm, while *Methyloligellaceae* and *Methylococcales* dominate the upper layer (0–2 cm and 4–6 cm). However, there is an overall lower relative abundance of methanotrophs across all depths in 2022 compared to 2021, with maximum values of approximately 0.9 % compared to 1.4 %.

A network interface of significant non-random relationships between the different hydrogenotrophic methanogenic groups and other microbes in the Lake Neusiedl sediment community was constructed (see Fig. 6). We obtained 1,398 nodes (ASVs)

and 132,305 edges, and after clustering and filtering, a co-occurrence network with 153 nodes and 325 edges were generated. In the network, the groups that displayed the highest number of relationships with methanogens included *Aminicenantales*, *Anaerolineales*, *Bacteroidales*, *Burkholderiales*, *Desulfatiglandales*, *Desulfobacterales*, *Pseudomonadales*, *Sedimentisphaerales*, and *Woesearchaeales*.

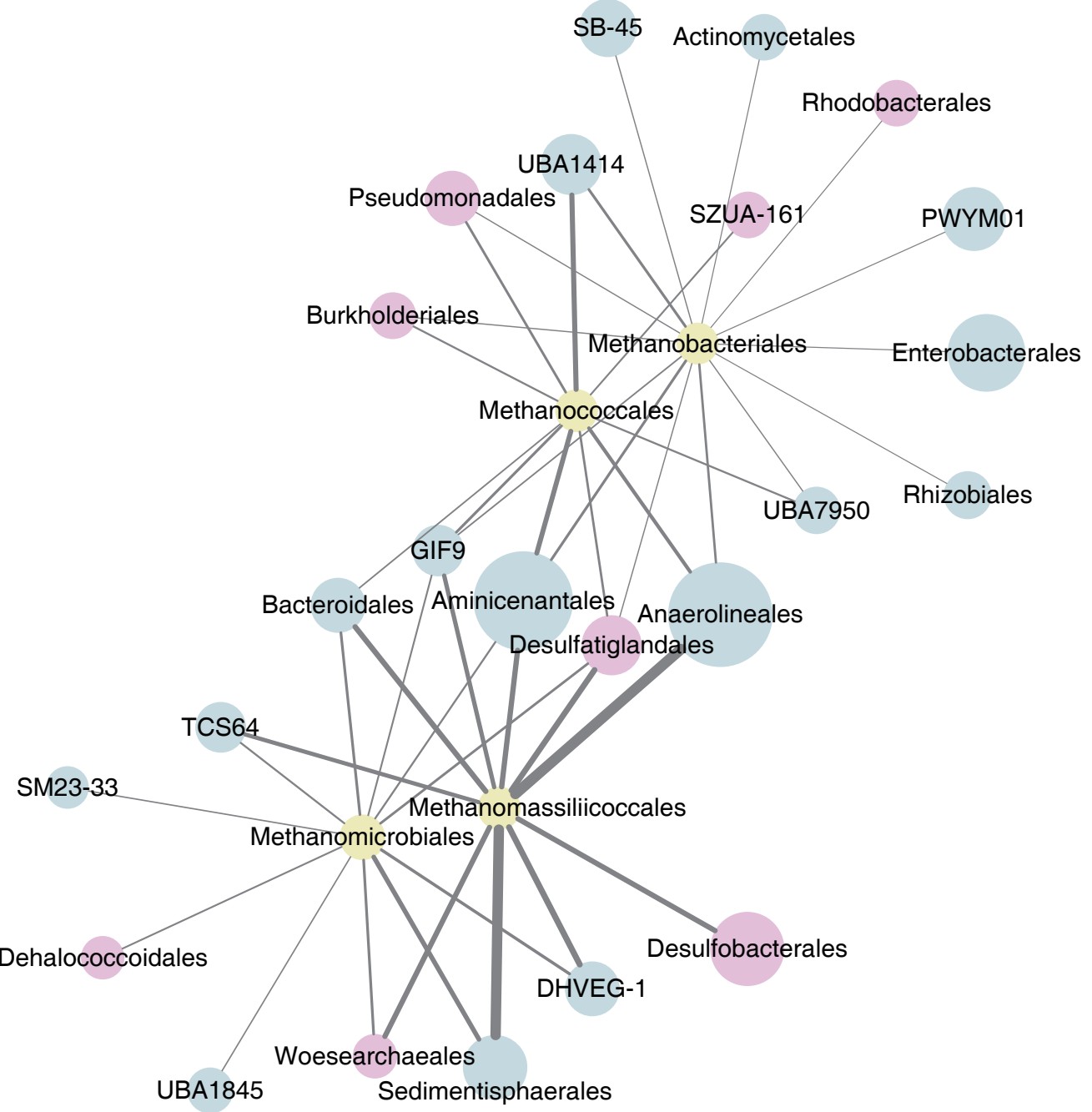

**Figure 6.** Co-occurrence network ($p < 0.05$) of hydrogenotrophic methanogens (yellow) with other microbes, with groups containing organisms with potential for acetate oxidation highlighted (pink). Only the top 10 microbes (blue) with links to methanogens were included. The size of each node is proportional to the relative abundance of the group and the thickness of each line is proportional to the number of links between the nodes.

## 4 Discussion

 ### 4.1 Seasonal and diel differences in $CH_4$ fluxes for each emission pathway in a subsaline reed wetland

This is one of the first studies to date to investigate all assessable $CH_4$ emission pathways in a wetland with reed over the entire diel cycle (24 h) and in all seasons individually. In the studied subsaline reed wetland, plant-mediated transport was identified as the most significant emission pathway for $CH_4$, irrespective of the season.

The highest seasonal median plant-mediated $CH_4$ flux (summer) at our studied wetland with reed is approximately half that of the average plant fluxes from studies of littoral zones in lakes or wetlands (8.3 mg $CH_4$ $m^{-2}$ $h^{-1}$), as reported by Bastviken et al. (2011). Moreover, our observed flux is 4 to 10 times lower than that reported in studies of other wetlands with reed (van den Berg et al., 2020; Duan et al., 2006) and twice as high as the flux observed in a subtropical *Phragmites* wetland subjected to drought (Jeffrey et al., 2019). One possible explanation for this discrepancy is the influence of lower water levels, given that Lake Neusiedl was experiencing drought (Baur et al., 2024b). A second possible explanation is that the subsaline reed ecosystem exhibited lower $CH_4$ production rates than the fen (van den Berg et al., 2020), which may be attributed to the coexistence and competition of sulfate reducers and methanogens.

The plant-mediated $CH_4$ fluxes at our site were consistently higher than diffusive $CH_4$ fluxes at the water–air and sediment–air interfaces, also in the non-growing season. One potential explanation is the chimney effect, which has been observed in the context of standing dead or non-intact reed culm between the sediment and the atmosphere (van den Berg et al., 2020; Brix, 1989). An additional potential explanation is Venturi-induced flow, as observed by Armstrong et al. (1992) in *P. australis*. This phenomenon is wind-induced and is likely to occur during the winter months when humidity-induced convection is absent.

In reed ecosystems, a clear diel cycle of $CH_4$ fluxes is only observed when pressurized flow is present, as opposed to diffusion. Pressurized flow is contingent upon the presence of living and intact reed culm (van den Berg et al., 2020; Vroom et al., 2022). Accordingly, at our site, a distinct diel cycle of $CH_4$ fluxes was discerned solely in summer and in instances of plant-mediated transport. However, when the reed ecosystem is considered as a whole and without partitioning in emission pathways, the reed belt exhibited a diel cycle with two peaks (with a slightly higher peak in the afternoon) during the summer months of 2021 (Baur et al., 2024b). This phenomenon is likely caused by $CH_4$ oxidation during daytime due to the low or absent water levels above the sediments (Baur et al., 2024b). The higher water level would also account for van den Berg et al. (2020)'s observation of a distinct diel cycle of $CH_4$ fluxes in June, both at the ecosystem level and in plant-mediated transport.

It can be reasonably assumed that the pressurized flow at our site is primarily driven by humidity and temperature. These findings are corroborated by observations that the plant-mediated $CH_4$ fluxes of *P. australis* exhibited a peak in the diel cycle in summer in the late afternoon, as did sediment temperature (in 5 cm depth) and VPD (see Fig. A1). In addition to light, these drivers were also identified by Vroom et al. (2022). In a reed wetland in an arid region of China, the diel cycle was explained by light-induced convection flow and the peak often occurred in the afternoon, coinciding with the time of the highest sediment temperature (Duan et al., 2006). In contrast, in a fen with reed, the peak of diel $CH_4$ fluxes were around midday (van den Berg et al., 2020), which could indicate the higher influence of light intensity and the lack of influence of soil temperature due to the high water levels. Another study conducted in the Hungarian part of Lake Neusiedl on 30 June 1993 revealed a daily maximum

of pressure and flow in the efflux reed culms in the afternoon (Brix et al., 1996), indicating a potential contribution of old reed stubbles to the afternoon peak of diel $CH_4$ fluxes. At our site, there are enough standing dead and broken reed culm in summer to potentially exert an influence.

The ebullition rates at our site are, on average, lower than the plant-mediated $CH_4$ emissions, with a notable increase observed in summer compared to spring. The median summer ebullition rate of 1.35 mg $CH_4$ m$^{-2}$ h$^{-1}$ is comparable to the average ebullition rates observed in a subtropical wetland (partially covered with reed) across two seasons (Jeffrey et al., 2019). However, at our study site, the release of ebullition was found to be highly sporadic and less frequent in spring. Consequently, ebullition measurements require the use of longer periods than 24 h outside of summer. Baur et al. (2024a) employed a sampling frequency of one week at Lake Neusiedl to more effectively capture the natural sporadic release of ebullition gases from April to July 2021. Nevertheless, the two 24 h campaigns with ebullition measurements adequately captured ebullition when compared to the median ($\pm$ standard deviation) of the continuously measured and weekly collected $CH_4$ ebullition rates within the reed belt, separated by seasons, in Baur et al. (2024a): In spring in Baur et al. (2024a), the median ebullition rate from March to May was $0.04 \pm 0.86$ mg $CH_4$ m$^{-2}$ h$^{-1}$, while the median $CH_4$ flux of the ebullition pathway in the spring campaign was $0.00 \pm 1.00$ mg $CH_4$ m$^{-2}$ h$^{-1}$ (see Table 2). During the summer season, when the water level was above the surface (from June to mid-July), the median ebullition rate was $1.19 \pm 1.35$ mg $CH_4$ m$^{-2}$ h$^{-1}$, while the median flux of the ebullition pathway in the summer campaign was $1.35 \pm 3.69$ mg $CH_4$ m$^{-2}$ h$^{-1}$. In winter, ebullition measurements with funnels were unfortunately not possible, however, ebullition is excepted to be very low in winter due to temperature dependence (Aben et al., 2017).

## 4.2 Seasonal methanogenic characterization with $\delta^{13}$C source signatures of a subsaline reed wetland

The present study demonstrated that a subsaline reed wetland exhibits seasonal variability in the source signatures $\delta^{13}$C-$CH_4$ and $\delta^{13}$C-$CO_2$. The most $^{13}$C-depleted $\delta^{13}$C-$CH_4$ source signature of $-73.6$ ‰ VPDB was observed in fall and may be attributed to the greater availability of substrates (acetate) at this time of year. In contrast, the most $^{13}$C-enriched $\delta^{13}$C-$CH_4$ source signatures were observed in spring and winter ($-59.2$ ‰ VPDB) and are likely due to reduced substrate availability (Sriskantharajah et al., 2012). Fisher et al. (2017) previously assumed seasonal variability in the isotopic source signatures of diel measurements in a Finish fen, with the most $^{13}$C-depleted $\delta^{13}$C-$CH_4$ in fall (October). However, their study did not cover all seasons.

Our summer $\delta^{13}$C-$CH_4$ source signature of $-62.6 \pm 2.4$ ‰ VPDB indicates a greater $^{13}$C-enrichment compared to the summer source signature of boreal wetlands, which has been determined to be $-70.9 \pm 1.2$ ‰ (Fisher et al., 2017). Whereas, the seasonal mean $\delta^{13}$C-$CH_4$ source signature of our site ($-63.6$ ‰ VPDB) is slightly more $^{13}$C-depleted than the ones from tropical reed wetlands with $-62.7$ ‰ (Hong Kong) or $-59.6$ ‰ (Zambia) (France et al., 2022). Fisher et al. (2017) elucidated the discrepancies observed in wetlands situated in more southern latitudes as a consequence of enhanced $CH_4$ oxidation in warmer wetlands, which exhibit thicker oxic layers, and due to variations in methanogenic communities. In our case, $CH_4$ oxidation was likely further facilitated by drought, resulting in a source signature comparable to that observed in tropical wetlands with reed.

This study corroborates the utility of Keeling plots derived from data of diel chamber measurements as a means of identifying source signatures of wetlands (Fisher et al., 2017). In summer, the $\delta^{13}$C-CH$_4$ source signature shows a high correspondence with the $\delta^{13}$C-CH$_4$ values of the collected ebullition gases. The source signatures illustrate, according to the methanogenic characterization plot of Whiticar et al. (1986), the predominance of acetoclastic methanogenesis across all seasons in the reed ecosystem under study. Until now, the dominance of acetoclastic methanogenesis at this site has only been confirmed during the April to July period by analyzing the ebullition gas (Baur et al., 2024a). However, the winter source signature indicates an influence of CH$_4$ oxidation due to the desiccation of the reed belt.

## 4.3 Sediment properties of a subsaline reed wetland

The reed wetland of Lake Neusiedl has two distinct sediment layers that differ in both appearance and properties. Despite the high TOC values in the upper sediment layer, the peaty layer is very shallow, with a mean thickness of $9.7 \pm 3.8$ cm. This is different from other reed wetlands, which are sometimes classified as minerotrophic peatlands due to their thicker peat deposits, such as the Federseemoor in van den Berg et al. (2020). In our studied reed ecosystem, most sediment properties, such as WC or sulfate, differ more between the two layers than between seasons. These properties show significantly higher average values in the upper, organic layer. However, the pH in the lower layer was significantly higher than in the upper organic layer. This difference may be due to the influence of organic or carbonic acids on the pH in the upper layer (Blume et al., 2016). A decrease in TOC and sulfate concentrations was also observed with sediment depth in a *Phragmites* wetland in Australia (Jeffrey et al., 2019). The lower sediment layer most likely has (strongly) reducing conditions throughout the year since alternative electron acceptors, such as nitrate and sulfate, are present in very low concentration or are not detectable. In the upper sediment layer, sulfate, the energetically less favored electron acceptor, is noticeably increased, suggesting less reducing (and more oxidizing) conditions. The sediment sample results also indicate spatial variability of the sediment properties, especially in the upper layer, which is in contact with the atmosphere or the surface water and where most processes, such as the decomposition of organic matter and the transformation of nutrients, occur.

The upper layer exhibited the highest seasonal variability in sediment properties, such as TOC, CON, and nutrients, which could be partly attributed to the particular drying conditions during the study period. In contrast, Agoston-Szabo and Dinka (2006) observed no temporal change in TOC in the upper 4 cm of sediment in the Hungarian reed belt of Lake Neusiedl from April to September 2001. Despite the drying out of the surface water areas ( = no water level above the sediment) in the reed belt from mid-July 2021 onward (after the summer campaign), the WC of the sediment in the upper (organic) layer remains relatively high due to saturation. However, since the sediment was no longer covered by water, and cracks had formed in it, the upper sediment layer (and perhaps deeper layer) had the opportunity for more direct contact with (atmospheric) oxygen. Additionally, sediment processes and properties may be influenced by the oxygen transport of *Phragmites australis* into the rhizosphere (Armstrong and Armstrong, 1988). The elevated ammonium concentration in the upper sediment layer in summer may result from freshly settled, easily degradable organic matter (van Luijn et al., 1999). The elevated nitrite concentration in the upper sediment layer in the following season (fall) could be explained by the oxidation of ammonium (1st step of

nitrification), presumably influenced by dry conditions. In the subsequent season (winter), the elevated nitrate concentration in the upper sediment layer could be explained by the oxidation of nitrite (2nd step of nitrification).

## 4.4 Change of the sediment microbial community in a subsaline reed ecosystem within one year

The variations observed in the methanogenic community of Lake Neusiedl between the two sampling years include differences in both relative abundance and diversity. Despite their higher relative abundance of methanogens, the overall microbial biomass in the deeper layer was consistently low across both years, as evidenced by the little DNA yield from these samples (see Table A1). *Methanomicrobiales*, an order detected in both 2021 and 2022, exhibited particularly high abundance in the latter year. Members of this group utilize $H_2$, $CO_2$, formate, and alcohols for methanogenesis (Garcia et al., 2006). *Methanococcales* and *Methanobacteriales*, also detected in 2021, encompass primarily hydrogenotrophic organisms (Bonin and Boone, 2006; Whitman and Jeanthon, 2006).

Notably, our campaign results indicate $CH_4$ production due to acetoclastic methanogenesis across all seasons. This process is exclusively carried out by representatives of *Methanosarcinales* (Stams et al., 2019), and while present in both years, these organisms were more prevalent in 2021 with lower levels detected in 2022. While it is worth pointing out that 16S rRNA gene relative abundance does not necessarily correlate to activity, these methanogens could be playing an important role in the $CH_4$ levels detected. Furthermore, syntrophic acetoclastic methanogenesis is known to occur, especially in environments less conducive to *Methanosarcinales* (Hattori, 2008). In this context, acetate is oxidized by organisms such as syntrophic acetate oxidizing bacteria, Woesearchaea, some sulfate-reducing bacteria, among others, and the resulting $H_2$ could be used for hydrogenotrophic methanogenesis (Lee and Zinder, 1988; Spormann and Thauer, 1988; Galushko and Kuever, 2019; Castelle et al., 2021; Huang et al., 2021). Interestingly, the co-occurrence network analysis (see Fig. 6) of sediment samples from Lake Neusiedl shows possible syntrophic interactions between hydrogenotrophic methanogenic orders and examples of such groups (highlighted in pink), indicating that syntrophic acetoclastic methanogenesis might be further contributing to the observed results. Moreover, the consistent detection of methanotrophs across all layer in both years could have affected the $CH_4$ levels measured in the present study, as these organisms have been shown to potentially consume a substantial amount of upward $CH_4$ fluxes in sediments (Valentine and Reeburgh, 2000).

The different levels of $O_2$ tolerance and sensitivity of methanogens may explain the differences in occurrences between the two sediment layers. Furthermore, the change in methanogenic taxa between the two winters (2021 and 2022) can be attributed to the drying of the study site from mid-July 2021, which resulted in the formation of cracks where the sediments could come into increased contact with $O_2$. First, we observed a (near) absence of methanogens in the upper layer in winter 2022 after desiccation, which could indicate that oxic conditions were dominant in this layer. Secondly, only the $O_2$-tolerant methanogens, such as for example *Methanosarcina subterranea* and *Methanosarcina siciliae* belonging to *Methanosarcinales* group (Conrad, 2020b; Lyu and Lu, 2018; Horne and Lessner, 2013; Angel et al., 2011), were present in the upper layer in winter 2021, when the study site was flooded. This indicates that the $O_2$ delivery to the rhizosphere may have occurred through the aerenchyma of *P. australis* (Armstrong et al., 1992), thereby influencing the $O_2$ content in this sediment layer, as standing water was above the sediments in winter 2021. Third, after desiccation, a marked increase and dominance of *Methanomicrobiales*, which are

$O_2$-tolerant methanogens (Conrad, 2020b; Lyu and Lu, 2018), was observed in the lower layer. Fourth, the lower layer prior to desiccation exhibited a high diversity of methanogens, including those belonging to the $O_2$ sensitive *Methanobacteriales* and *Methanococcales* (Conrad, 2020b; Lyu and Lu, 2018). However, these were absent after desiccation. These observed changes are inconsistent with the findings of Conrad (2020b), who was able to find methanogenic population dynamics after desiccation, but not with $O_2$ sensitivity as the exclusive criterion.

Considering that the sulfate reduction zone typically occurs closer to the surface compared to the methanogenesis zone (Jørgensen and Kasten, 2006), the observed higher relative abundance of sulfate-reducing bacteria in the upper layer and methanogens in the lower layer is not surprising. Furthermore, the increased detection of *Desulfovibrionales* with depth could be a consequence of a decrease in organic matter in the deeper layer (see Fig. 4a), as members of this group have been shown to thrive in oligotrophic environments (Sass et al., 1998; Bade et al., 2000; Wörner and Pester, 2019). The detection of sulfate-reducing bacteria across both years draws attention due to their metabolic features in aquatic sediments. Acetate and $H_2$ serve as substrates for both sulfate reduction and methanogenesis (Schink, 1997; Conrad, 1999). Sulfate-reducing bacteria hold a thermodynamic advantage over methanogens, as they can utilize acetate and $H_2$ at lower concentrations, potentially outcompeting methanogens for substrate uptake. This process channels electron flow towards $CO_2$ production rather than $CH_4$ (Lovley et al., 1982; King, 1984; Lovley and Goodwin, 1988). However, it is worth noting that these organisms and processes can co-occur, as observed in environments such as the sulfate-methane transition zone, and could be coupled in ways that influence substrate production and consumption rates (Egger et al., 2016; Sela-Adler et al., 2017). Thus, it is possible that similar interactions occur in Lake Neusiedl. The higher relative abundance of sulfate reducers in the upper layer, followed by a decrease in their levels in the lower layer coinciding with an increase in methanogen relative abundance, suggests a putative competition scenario where methanogens may have been outcompeted for resources when sulfate reducers were more prevalent.

## 5 Conclusions

This study shows that in a wetland with reed, a distinct diel cycle of $CH_4$ fluxes occurs only in the emission pathway of plant-mediated transport and in the summer season. The plant-mediated transport pathway demonstrated the highest $CH_4$ emissions not only in summer, but also during other seasons. The distinct differences in the two sediment layers, namely in terms of carbon and water content, are also reflected in the variations observed in the microbial communities. The desiccation process resulted in a reduction in the methanogenic microbial diversity in the sediments over the course of a year. The findings of this study can be extended to other wetlands with reed vegetation, particularly those situated in similar subsaline or brackish conditions, or to other wetlands experiencing drought conditions. However, additional research is required to determine whether the methanogenic diversity will recover following an increase in water levels in reed wetlands.

## A1    Diel and seasonal differences in biometeorological and water parameters

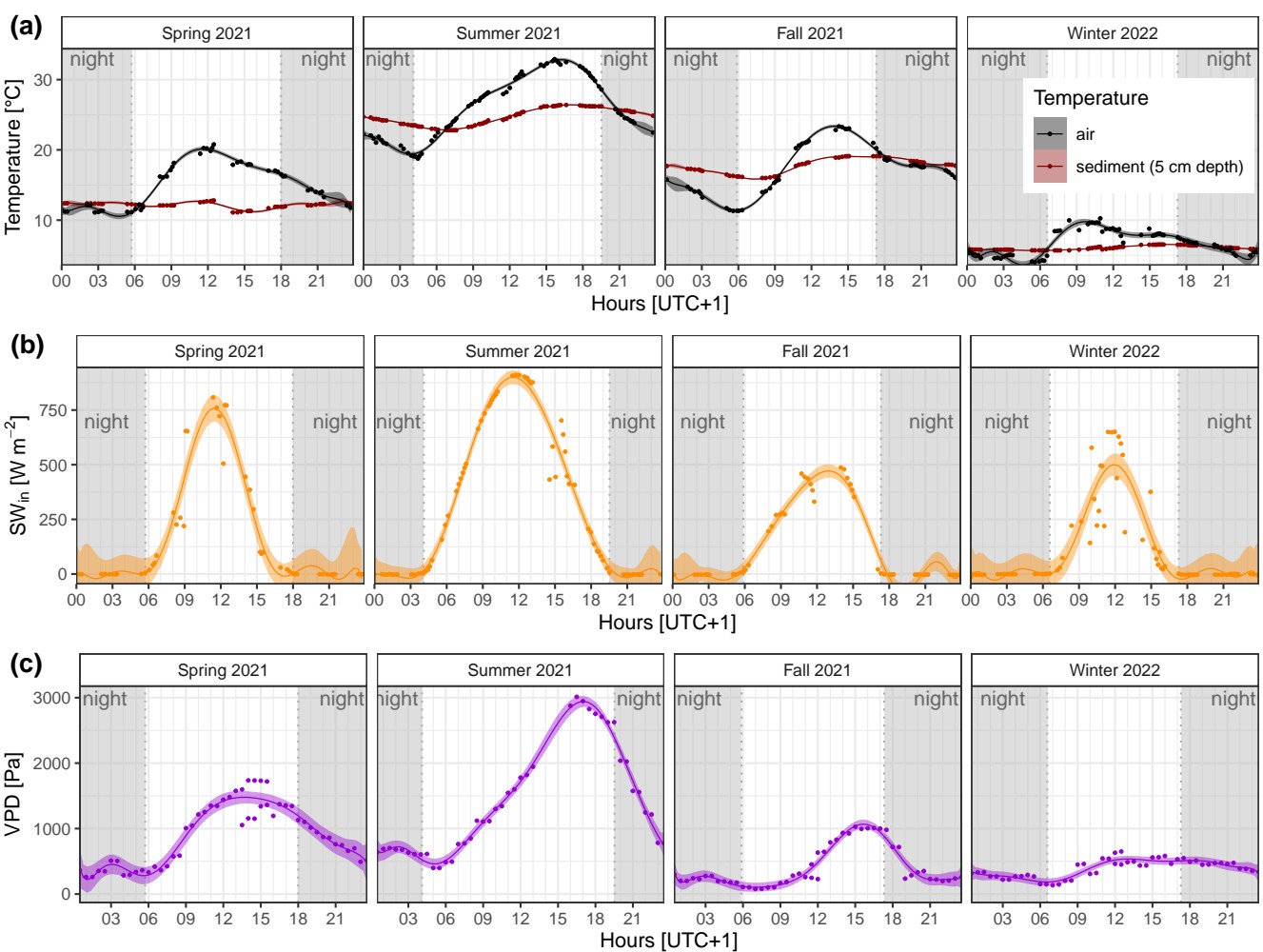

**Figure A1.** Diel and seasonal differences in **(a)** ambient air and sediment temperature, **(b)** incoming shortwave radiation (SW$_{in}$), and **(c)** water vapor pressure deficit (VPD) in the reed belt of Lake Neusiedl (data source: Baur (2024)); the fitted lines showing the decic polynomial regressions with the 95 % confidence interval; nighttime when SW$_{in}$ was < 10 W m$^{-2}$ (gray shading).

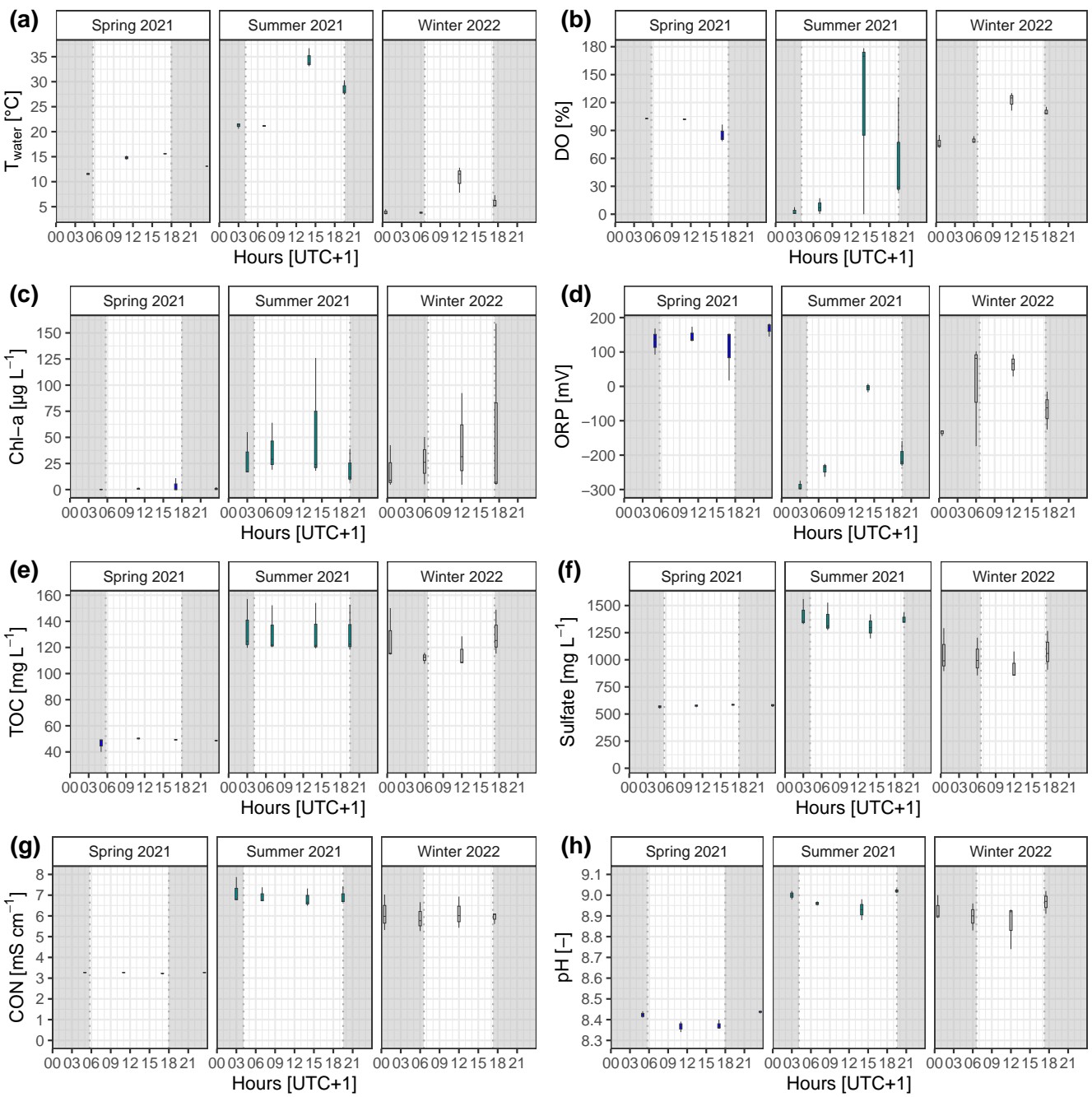

**Figure A2.** Diel and seasonal variability in surface water properties of the reed belt during 24 h campaigns: **(a)** water temperature ($T_{water}$), **(b)** dissolved oxygen (DO), **(c)** chlorophyll-a concentration (Chl-a), **(d)** oxygen–reduction potential (ORP), **(e)** total organic carbon concentration (TOC), **(f)** sulfate concentration, **(g)** electrical conductivity (CON), and **(h)** pH. Nighttime when incoming shortwave radiation was < 10 W m$^{-2}$ (gray shading). In fall 2021 no water above the sediment surface was present.

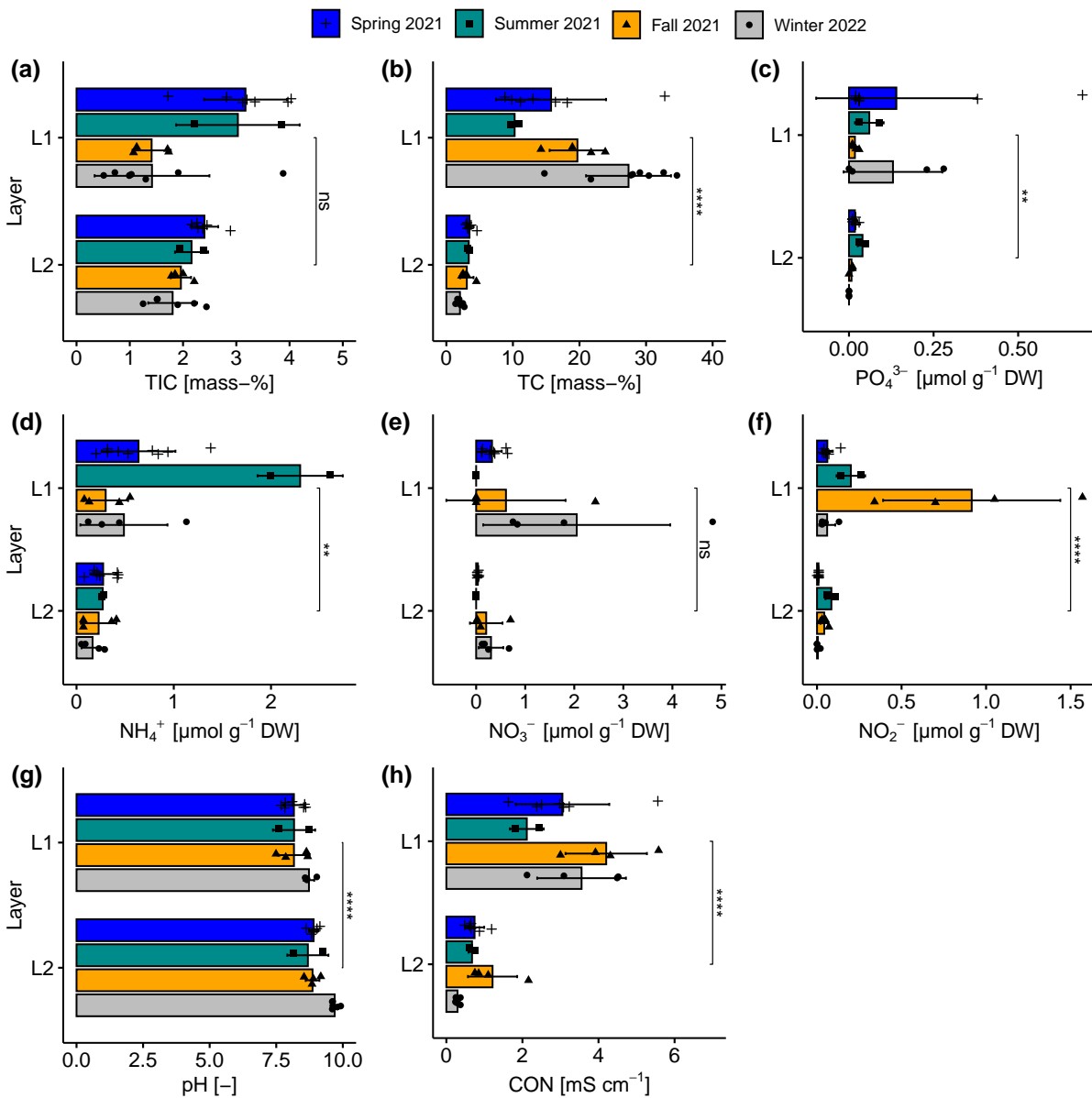

**Figure A3.** Seasonal and layer specific differences of **(a)** total inorganic carbon content (TIC), **(b)** total carbon content (TC), **(c)** phosphate ($PO_4^{3-}$), **(d)** ammonium ($NH_4^+$), **(e)** nitrate ($NO_3^-$), **(f)** nitrite ($NO_2^-$) concentration, **(g)** pH, and **(h)** electrical conductivity (CON) in the upper (L1) and lower layer (L2) of sediments in the reed belt of Lake Neusiedl. Non-significant (ns, $p > 0.05$ with Holm's adjustment, Dunn's test), significant (**, $p < 0.01$), or very significant differences (****, $p < 0.0001$) in the parameters between the two layers in the sediments independent of the season.

## A3 Inter-annual differences in properties and microbial characteristics of sediments

**Table A1.** DNA yield, total number of microbial classes, gravimetric water content (WC), electrical conductivity (CON), and total organic carbon content (TOC) in four depth section of the sediment cores I and II from the reed belt of Lake Neusiedl in winter 2021 and 2022. The two sediment sections, 0–2 cm and 4–6 cm, are part of the upper layer (L1) of the sediment, while the sections 20–22 cm and 24–26 cm are located in the lower layer (L2).

| Season | Core | Depth section [cm] | DNA yield [$\mu$g g$^{-1}$ DW] | Number of classes | TOC [mass-%] | WC [mass-%] | pH | CON [mS cm$^{-1}$] |
|---|---|---|---|---|---|---|---|---|
| **Winter 2021** | I | 0–2 | 41.172 | 143 | 33.2 | 89.1 | 8.36 | 3.35 |
| | | 4–6 | 39.549 | 146 | 35.1 | 89.6 | | |
| | | 20–22 | 0.033 | 112 | 0.5 | 24.8 | 9.13 | 0.34 |
| | | 24–26 | 0.042 | 124 | 0.8 | 31.0 | 8.76 | 0.88 |
| | II | 0–2 | 43.571 | 162 | 31.5 | 90.8 | 8.12 | 2.25 |
| | | 4–6 | 42.747 | 166 | 30.9 | 89.6 | 8.33 | 2.56 |
| | | 20–22 | 0.037 | 127 | 0.3 | 26.3 | 9.10 | 0.35 |
| | | 24–26 | 0.001 | 84 | 0.2 | 21.4 | 9.34 | 0.35 |
| **Winter 2022** | I | 0–2 | 8.058 | 141 | 29.4 | 87.8 | 8.65 | 4.49 |
| | | 4–6 | 9.889 | 134 | 31.9 | 87.8 | | |
| | | 20–22 | 0.628 | 108 | 0.4 | 24.0 | 9.60 | 0.37 |
| | | 24–26 | 0.300 | 117 | 0.3 | 21.4 | 9.59 | 0.32 |
| | II | 0–2 | 7.899 | 127 | 26.8 | 88.0 | 8.65 | 4.52 |
| | | 4–6 | 7.844 | | 26.7 | 89.1 | | |
| | | 20–22 | 0.558 | 158 | 0.1 | 19.4 | 9.62 | 0.25 |
| | | 24–26 | 0.282 | 106 | 0.3 | 22.6 | 9.53 | 0.33 |

*Data availability.* The sediment, isotope, and flux data generated by this study are available in the Phaidra repository of the University of Vienna under https://doi.org/10.25365/phaidra.626. All 16S rRNA raw sequences data are available at the NCBI database under the project identifier PRJNA1182700.

*Author contributions.* PAB: conceptualization, data curation, formal analysis, investigation, methodology, project administration, software, validation, visualization, writing - original draft preparation, writing - review & editing; TRO: conceptualization, data curation, formal analysis, methodology, writing - original draft preparation, writing - review & editing; KH: data curation, formal analysis, methodology; ZHL: data curation, formal analysis, visualization, writing - review & editing; CS: funding acquisition, resources, supervision; SG: funding acquisition, resources, supervision, writing - review & editing.

*Competing interests.* The authors declare no conflict of interest.

*Acknowledgements.* We highly acknowledge the intensive field and lab support of the Geoecology team and others: Claudia Buchsteiner, Daniela Henry, Andreas Maier, Raphael Müller, Sebastian Echeverria, Camila Aguetoni Cambui, Katharina Fischer, Yujing Deng. We thank the team of Thomas Zechmeister from the Biological Station Lake Neusiedl (Illmitz, Austria) for their support in the construction of the boardwalk and water analyses in the lab. We also thank the national park 'Neusiedler See – Seewinkel' for access to the study area, the
granting of permits and the friendly cooperation. We would like to thank the Burgenland Provincial Government for granting the permit ('Naturschutzbehördliche Ausnahmegenehmigung') for entering the national park nature zone and conducting research there.

*Financial support.* This research was funded in part by the Austrian Science Fund (FWF) [10.55776/Z437] and in part by ERC AdvGrant TACKLE (695192).

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
