# Peer review of "Temporal dynamics of CH4 emission pathways in the subsaline reed wetland of Lake Neusiedl"

_EGUsphere, 2025_

## Author Comment (AC1)

**Discussion of "Pathways of CH4 formation and emission in the subsaline reed wetland of Lake Neusiedl"**

**Baur et al.**

In the following, the reviewer's comments are set in *italic* and the authors' responses are marked in blue.

**Authors' response to Referee 1**
* * *
*Review of "Pathways of CH4 formation and emission in the subsaline reed wetland of Lake Neusiedl"*

We thank the reviewer for the time and the valuable comments for improving our manuscript.

*Here, Baur et al. continued the extensive research previously conducted by this scientific group in a reed wetland located at a lake in Austria, the title argue that they were focusing on understanding CH4 formation pathways. Although the title attempts to encompass this complex objective, I believe the methodology employed does not fully address such broad scenarios. Instead, I recommend emphasizing diel variations across different seasons rather than the overall CH4 formation processes in the studied ecosystem. I mention this because you are inferring CH4 pathways without performing an appropriate experimental analysis. Microbial assessments combined with isotopic measurements from emissions alone might not be sufficient to clearly identify these pathways. For robust conclusions, incubation experiments using substrates or more detailed analyses would be necessary. I am currently providing general comments on the manuscript, in the next review round, I will likely offer more detailed feedback. At this stage, however, I think the manuscript needs substantial improvement.*

We thank the reviewer for the comment and agree that the title was perhaps a bit too broad and that not everything could be fully answered with this study. However, we wanted to provide insights into the various emission pathways, seasonal isotopic source signatures, methanogenic characterization (following Whiticar et al. (1986)), and the microbial communities involved in this ecosystem. We suggest changing the title to "Temporal dynamics of CH4 emission pathways in the subsaline reed wetland of Lake Neusiedl".

*Firstly, the Introduction appears poorly organized. Although I understand you might be addressing topics similar to those from previous manuscripts, here the Introduction is brief and omits important details. Regarding isotope information, please ensure that you clearly describe and contextualize isotopic methods and their relevance in terms of diel variations, and their respective emission. Additionally, having paragraphs consisting of only one sentence is inadequate, please expand or combine such sentences with relevant information to provide greater clarity. Several process definitions are merely mentioned but insufficiently explained. For example, you need to clearly introduce the effects of sulfate and oxygen availability on sediments, as well as their potential impacts on CH4 emissions. Not all variations in CH4 emissions can be explained by oxidation alone; rather, the microbial community may also utilize alternative substrates for respiration. This aspect is currently missing and should be addressed explicitly in the Introduction.*

Thank you for your recommendations. We will improve the Introduction chapter by expanding the explanations of the various processes in the sediments involved, including the isotopic methods, and by reorganizing the chapter structure. However, we must mention that the diel variability of isotope values (e.g. from gas in the atmosphere) was not part of this study, only the seasonal change of the isotopic source signature of $CH_4$ (in the sediments).

*Secondly, I understand that the methodology was adopted from previous studies, however, you need to briefly include specific details about your own methods. For example:*

*(i) Please include, either as an appendix or supplementary material, information about the observed flux determined using the chambers and the Picarro analyzer, clearly indicating instances when data were too erratic. Additionally, specify the Picarro measurement mode and the frequency of data acquisition. For isotope analyses, clarify how long measurements were stabilized before recording values, as well as the measurement uncertainties (I remember it is necessary like 6 min to confirm the isotopic value).*

Yes, these details are mentioned in Baur et al. (2024b), but for better traceability, we will add more information in the appendix, e.g. about the Picarro measurement mode, Picarro measurement frequency, isotope analysis, and flux equations/determination.

*(ii) Table 1 currently provides limited information; please add the sample sizes corresponding to the reported uncertainties. Lastly, regarding the laboratory-analyzed samples (although not clearly described), could you clarify if the reported standard deviations for sulfate, TOC, and chlorophyll-a resulted from temporal sampling conducted over a 24-hour period?*

Yes, most parameters in Table 1 show the temporal means and standard deviation over 24 h, but it also includes 3 spatial replicates (see the 3 water sample locations in Figure 1d). Only for two parameters (water level and leaf area index), which are not assumed to change over 24 h, only spatial means and standard deviation are given. We will make this clearer by including this information in the caption of Table 1 and also add the sample sizes. This table is located in the subchapter 2.2 "Measurement setup" of the Methods chapter and describes the environmental conditions at the site during each 24 h campaign. The intention of this table was also to show and allow their comparability with other days/nights in the same season. In order to keep the manuscript concise, we have refrained from including the water sample data in the results chapter. However, as the reviewer question indicates, that this might be of great interest for the reader, so we will add the lab methods (of water analysis) and detailed data, including diel variability of these parameters, e.g. in the appendix.

*Also, I am curious about how you obtained ebullition data during periods of very low water levels, particularly in summer when this emission pathway appears most prominent in your results.*

In the Baur et al. (2024b) study, the bubble traps were permanently installed from March to July 2021 until the water level in the reed belt was no longer above the surface. As the traps were installed at a time of higher water levels (early March), we were able to measure ebullition (collection of trapped gases) with these traps at time of very low water levels (June/July). However, for our study here, the gas collection interval of the bubble traps was every 6 h to investigate if there was a diel pattern, to quantify the ebullition pathway during 24 h and also to compare it with the other pathways. For the rest of the time, the trapped gas was collected once a week and used for the study of Baur et al. (2024b). These results show an increase in the ebullition rate in the reed belt from March to July. Contrary to your comment, we do not see

the ebullition pathway as the most prominent emission pathway in summer due to two high flux rates (out of 36), but rather the plant-mediated transport due to the maximum and highest median flux rate (see Table 2 and Figure 2).

*Furthermore, I am surprised by the high chlorophyll-a values observed in winter, are you certain these measurements represent chlorophyll-a and not turbidity caused by shallow water levels?*

In the reed belt, the surface water has a brownish color, but it is clear and not turbid (unlike the open lake water of Lake Neusiedl). The relatively high chlorophyll-a concentrations in winter are probably more likely due to algae and low water levels (higher concentration per liter).

*(iii) Ebullition measurements using bubble traps appear highly biased due to the short deployment duration. Given that ebullition is a very stochastic process, it is possible that significant emissions were missed simply because the traps were placed away from emission hotspots. I understand the difficulty involved, as this is one of the major challenges when studying ebullition. In previous studies, you deployed bubble traps for longer periods, however, the focus on diel variations in this study limited that possibility. Do you think a single measurement day per season provides sufficient data to establish a representative ebullition pattern? Looking into the data I would not discard as the main source of emission.*

Since we had 9 bubble traps (always 3 traps on one board) collecting bubbles in the reed belt, we believe that the spatial heterogeneity of ebullition and ebullition hotspots are well reflected. In order to investigate the diel pattern in different seasons in this study, the gas collection time of the bubble trap had to be shorter. Combined with the other study on ebullition rates at different sites and over a longer period of time (Baur et al., 2024b), we have a better picture of ebullition at Lake Neusiedl. Of course, it is always better to have more measurement days, but since 24 h campaigns are very resource intensive, we were not able to conduct more such campaigns. In addition, the drying out of the reed belt was not foreseen, otherwise longer measurements of the bubble traps would have been possible (after July).

*The soil chambers faced several issues related to ebullition (as you commented in the method section), suggesting that significant amounts of CH4 accumulated in the surface sediments,which may have been missed due to the short deployment time of the bubble traps (in this case absence of water, but still emission). Given this limitation, why did you not consider using the ebullition events detected by the Picarro analyzer for fluxes to determine other sources of CH4? With proper data management, you could potentially quantify ebullition as part of the total CH4 flux using these measurements.*

Yes, in the summer campaign there were ebullitions at the beginning or during chamber closure in 8 out of 24 sediment chamber measurements. These ebullitions were likely triggered by the movement of the chamber closure, because the top sediment layer ("fluffy layer") was very mobile, as mentioned in the Methods chapter in L160ff. Thus, we consider this release not representative. Since one aim of the study was to investigate the diel pattern of the naturally released methane emissions for each pathway and season (and not the total ecosystem emissions, which were investigated in Baur et al. (2024a)), we refrained from using the possibly artificially triggered ebullitions at the beginning or during the sediment chamber closure. As the experimental design has foreseen to measure the diffusion pathway with sediment chambers and the ebullition pathway at the same time (in summer) with the bubble traps, we did not want to overestimate this pathway by double counting. Since the bubble traps collected a lot of

bubbles during the summer campaign, we believe that we are taking the accumulation of methane in the top sediment layer into account, at least to a certain extent.

*Thirdly, the presented results represent measurements performed over just 1 day per season. I understand this limitation, it is because you were also collecting data on additional topics, which have been integrated into other published manuscripts. Although you made a substantial effort to collect this data, some results may still be significantly biased due to the short sampling period, particularly for ebullition as previously mentioned.*

Yes, only one 24 h campaign was conducted per season. This may lead to some limitations, but we also had to consider the feasibility and the purpose of this study. We still believe that this study contributes to understand the diel variations in this ecosystem and its emission pathways, despite only one year of measurements. In addition, we have published a study with continuous eddy covariance measurements that provide information on the total methane fluxes of this ecosystem (see Baur et al. (2024a)). As mentioned in L362ff, it makes sense to extend the ebullition measurement time intervals, especially in spring. However, since higher ebullition rates are associated with higher temperatures, e.g. in summer, shorter measurement time intervals of 6 h are also useful, as the measured values show (67% of the ebullition rates greater than zero, see L247ff). Since the focus of this study is on the different emission pathways, we wanted to compare the ebullition rate with the other emission rates and their diel pattern.

*But also, plant-mediated emissions are not fully isolated, as total emissions originating from sources external to the plants could have contributed.*

We are not sure if we understood your comment the way you meant. Here are our responses on how the emissions from the vegetation chambers or the plant-mediated transport itself might have been affected.

We agree that for vegetation chambers, the plant-mediated transport of reeds within the chambers cannot be completely isolated from diffusion from the underlying sediment or water surface. However, the major contribution or dominance of plant-mediated transport in the emission rates in the vegetation chamber compared to the diffusion pathway (at the water-air or sediment-air interfaces) is quite clear, as can be clearly seen from the much higher emission rates from the vegetation chambers in each season compared to the sediment or floating chambers. Therefore, we believe that the effect on the total emissions in the vegetation chambers should be relatively small. Furthermore, vegetation chambers are a common method for estimating plant-mediated transport of reed plants, see also van den Berg et al. (2020) or Jeffrey et al. (2019).

We also agree that the reed stems are probably well connected to each other by their rhizomes and are most likely all form "one" plant. However, this makes it almost impossible to isolate one reed stem by any measures (e.g. clipping, etc.) without destroying the reed ecosystem in that area. We had the opportunity to conduct research in the core zone of a national park with an exceptional permit and were therefore instructed to disturb the ecosystem as little as possible. We kept the chamber closure time as short as possible to minimize artefacts. Due to the installation of the bottom frame of the chambers in advance (in the winter before the campaigns), we had to cut the sediment at least 10 cm deep, which of course also meant that the rhizome connections with the surrounding area might have been severed.

*The figures require additional information, as it is currently difficult to interpret the origin of the presented data. For instance, in Figure 2, it's unclear what each data point represent, are*

*these individual measurements or average? If they represent average values, please clearly indicate this and the uncertainties. Additionally, if a polynomial function has been applied, specify its order. However, I question whether using a polynomial fit is appropriate in this context; please clearly justify its application as a fit rather than presenting the raw temporal data directly. Also what is the point to fit to this polynomial?*

In Figure 2, each point represents a single flux rate. We will make this clear in the caption and also indicate the order of the polynomial function. Since the data points represent the raw flux time series, we used the polynomial fit (with the 95% confidence interval) to visualize the average (and uncertainty) diel pattern of each emission pathway, including the peak (if available). An advantage of a polynomial fit is that it can also show smaller temporal variations, and in our case, it gave us the opportunity to show the temporal pattern with the raw data points and the exact measurement time without aggregating over time (e.g. one measurement round). We consider this advantageous as for example, in summer we had to measure 3 different manual chamber types with 3 spatial replicates each, it took quite some time to measure all of them one after the other (including flushing time of the cable and the measuring cell of the gas analyzer) and therefore the light or other meteorological conditions could have changed during one measurement round.

*Regarding Figure 3, the source of the data is unclear, and I am particularly interested in understanding why these results differ substantially from those previously published in your earlier manuscript in Baur et al. (2024, DOI: 10.1016/j.scitotenv.2023.169112). There is no clear oxidation pattern in Figure C. So you need to expand the information about it. Does this figure include all isotopic signature data from Figures A and B in the four points? Additionally, could you clarify at what value is used during the Picarro measurements?*

The data points shown in Figures 3a and b (Keeling plots) are in the x axis the reciprocal of the mean dry mole fraction of $CH_4$ (or $CO_2$) and in the y axis the mean stable carbon isotope ratio $\delta^{13}C$ of $CH_4$ (or $CO_2$), measured with the Picarro G2201-i during the closure of each chamber measurement. These points are used to calculate the seasonal isotopic source signatures (y-intercept of the regression line) using the Keeling plot approach. This approach is explained in the Methods chapter in L118ff and L125ff. However, we will clarify this with a better description in the caption of Figure 3.

The methanogenic characterization plot after Whiticar et al. (1986) in Figure 3c uses the four calculated seasonal isotopic source signature pairs, which were determined in Figures 3a and b as the y-intercept values of the regression lines and their standard errors. The winter source signature with its uncertainty (standard error) colored in black in Figure 3c is partly in the range of methane oxidation. This means that in this plot the carbon fractionation factor $\alpha_c$ is between 1.005 and 1.03. To facilitate understanding of the values used in Figure 3c, we will adjust the explanation in the caption.

In Baur et al. (2024b), in contrast to the manuscript submitted here, the directly measured $\delta^{13}C$-$CH_4$ values of the collected gas from the bubble traps are used in the methanogenic characterization plot. Ebullition is known to have no isotopic C fractionation, unlike other processes such as diffusion or methane oxidation. Therefore, the isotopic values of the ebullition gases can be used directly as an indicator of the methane source in the sediments without calculating the source signatures with a Keeling plot, as it was done here for all chamber measurements. Our study confirms the results of the methanogenic characterization plot (according to Whiticar et al. (1986)) from the previous study that acetoclastic activity is indicated as the main pathway of methanogenesis in this reed ecosystem, not only for the

months of March to July (Baur et al., 2024b), but for all four seasons (see Figure 3c). In addition, this study was the first to determine the seasonal source signatures for $CH_4$ and $CO_2$ for all four seasons in a reed wetland (y-intercepts in Figures 3a and b).

We will make these points clearer in the revised manuscript.

**References**

Baur, P. A., Maier, A., Buchsteiner, C., Zechmeister, T., and Glatzel, S.: Consequences of intense drought on CO2 and CH4 fluxes of the reed ecosystem at Lake Neusiedl, Environ. Res., 262, 119907, https://doi.org/10.1016/j.envres.2024.119907, 2024a.

Baur, P. A., Henry Pinilla, D., and Glatzel, S.: Is ebullition or diffusion more important as methane emission pathway in a shallow subsaline lake?, Sci. Total Environ., 912, 169112, https://doi.org/10.1016/j.scitotenv.2023.169112, 2024b.

Jeffrey, L. C., Maher, D. T., Johnston, S. G., Maguire, K., Steven, A. D. L., and Tait, D. R.: Rhizosphere to the atmosphere: contrasting methane pathways, fluxes, and geochemical drivers across the terrestrial–aquatic wetland boundary, Biogeosciences, 16, 1799–1815, https://doi.org/10.5194/bg-16-1799-2019, 2019.

van den Berg, M., van den Elzen, E., Ingwersen, J., Kosten, S., Lamers, L. P. M., and Streck, T.: Contribution of plant-induced pressurized flow to CH4 emission from a Phragmites fen, Sci. Rep., 10, 12304, https://doi.org/10.1038/s41598-020-69034-7, 2020.

Whiticar, M., Faber, E., and Schoell, M.: Biogenic methane formation in marine and freshwater environments: CO2 reduction vs. acetate fermentation—Isotope evidence, Geochim. Cosmochim. Acta, 50, 693–709, https://doi.org/10.1016/0016-7037(86)90346-7, 1986.

---

## Author Comment (AC2)

**Discussion of "Pathways of CH₄ formation and emission in the subsaline reed wetland of Lake Neusiedl"**

**Baur et al.**

In the following, the reviewer's comments are set in *italic* and the authors' responses are marked in blue.

**Authors' response to Referee 2**

We thank the reviewer for the time and the valuable comments for improving our manuscript.

*The authors assessed four pathways of CH4 fluxes (vegetation-mediated, soil- and water-air diffusion, and ebullition), pathways of methanogenesis (acetoclastic, hydrogenotrophic, and methyltrophic), microbial communities, and sediment physical and chemical properties in a subsaline reed-dominated temperate wetland. CH4 fluxes were measured over 24-hour periods 4 times over one year covering all four seasons (spring, summer, fall, winter). Only vegetation-mediated CH4 flux was measured in all 4 seasons.*

Yes, that is correct. Due to the intra-annual dynamics in this wetland with reed, the available land cover (water areas, open sediment areas, and reed stands) and thus the corresponding interfaces and emission pathways are not always available in every season (Buchsteiner et al., 2023). For example, in spring, the water level was high, so there was no sediment-air interface. In fall, however, the reed wetland was dry (no water level above the surface), so there was no water-air interface.

*The abstract implies that the research results contribute to reducing uncertainty in estimates of CH4 emissions from wetlands and improving wetland greenhouse gas modeling. However, the link between the research results and their significance needs clarification in the main text. Specifically, how can the results improve models, and what important knowledge gaps do they fill?*

In the revised manuscript we will more clearly describe the knowledge gaps in currently applied wetland CH₄ emission models and how our results can significantly contribute to reducing uncertainty in these models.

The CH₄ emissions in process-based vegetation models, e.g. like LPJ-GUESS in Kallingal et al. (2024) or CLM4Me model in Riley et al. (2011), only partially implement the plant-mediated transport by considering only the passive mechanism (concentration gradient) (Riley et al., 2011; Kallingal et al., 2024). They also do not consider the pressurized flow that occurs in plant species such as *Phragmites* or *Typha* sp. and others, although it is known that, for example, the CH₄ emissions by plant-mediated transport of *Phragmites australis* can be more than five times higher than by diffusion (Brix et al., 2001). According to Vroom et al. (2022), most other wetland greenhouse gas models do not implement plant-mediated CH₄ fluxes in wetlands and do not include them in the total flux due to high variability in their contribution and the lack of data about this variability.

- ➢ Therefore, our study can help to fill important knowledge gaps
  - o by quantifying the temporal contribution (seasonal and diel) of plant-mediated transport of reed wetlands and highlighting the dominance of plant-mediated transport not only in summer (as previous studies did, e.g. van den Berg et al. (2020)).
  - o by better understanding these emission pathways and their temporal (diel) pattern of the studied reed wetland for each season
  - o by using the findings to improve the parametrization and implementation of emission pathways (especially the plant-mediated transport of wetland plants) in the models.
- ➢ Furthermore, the study contributes to alleviating the general lack of plant-mediated $CH_4$ flux data in wetlands, so that plant-mediated transport and especially the pressurized flow mechanism will hopefully be implemented in (more) wetland $CH_4$ emission models to reduce their uncertainties, but also to make them more realistic.
- ➢ In addition, our $CH_4$ flux data (for each available emission pathway) can be used to optimize wetland $CH_4$ emission models as validation data for modeled $CH_4$ emissions for plant-mediated transport, ebullition, and diffusion in process-based vegetation models.
- ➢ Moreover, the findings of this study contribute to a more comprehensive understanding of the characteristics of reed wetlands, including the temporal $CH_4$ emission dynamics, their sediment and water properties, and their microbial communities.

*Discussion of the results of sediment physical and chemical property analysis is absent and should be added.*

We will add a paragraph in the Discussion chapter, in which we will discuss the results of the physiochemical analysis of the sediments in the reed wetland, their seasonal changes and differences compared to other (reed) wetlands.

*The authors should also define the term "subsaline" and clarify the significance of the wetland type studied. This is particularly important for convincing non-specialists of the research impact.*

According to Hammer (1986), subsaline lakes are defined as lakes with a salinity between 0.5 and 3‰. We will include this salinity range in front of the cited reference in L56.

The significance of the wetland type studied can be demonstrated for several reasons, which we will clarify in the revised manuscript:

The studied wetland type is one of the largest connected reed wetlands in Europe and has a special water characteristic for inland waters, e.g. subsalinity. Subsaline wetland ecosystems, and especially subsaline reed ecosystems, are rarely ever studied and therefore need to be (further) investigated. Because of the salinity, these ecosystems are to some degree more comparable to brackish coastal ecosystems than to freshwater ecosystems. However, the salt composition of the reed wetland in this study differs from that of coastal salt marshes because the main salt is sodium bicarbonate, not sodium chloride. It also differs greatly from other inland ecosystems such as reed fens (see e.g. van den Berg et al. (2020)) due to different soil properties. In addition, the studied lake, in which more than half is covered by the reed wetland, has been severely affected by droughts since mid-2015, which have influenced the carbon cycle, its processes and fluxes (Baur et al., 2024a). The international importance of this wetland is also shown by its cross-border protected areas as a Ramsar site, a UNESCO World Heritage

site, and a National Park. All these made it very interesting to study this wetland type for a better understanding of carbon-related processes in different wetland types and conditions.

*I also have some concerns regarding the research design using punctual 24-hour assessments of fluxes over four seasons, which may not adequately capture within-season temporal variation in fluxes, particularly ebullition. Further justification of 24-hour monitoring periods and comparison of flux pathways across seasons is needed. Is there potential for bias in the seasonal measurements given the short duration of monitoring periods?*

*If the authors are willing to address these issues, I'm happy to review a revised version of the manuscript in further detail.*

We will address your questions in the revised manuscript:

We have taken great care to come up with meaningful results by studying the seasonal differences in the diel variation of the $CH_4$ fluxes for each available emission pathway of a reed wetland. We used the research design of one intensive 24 h campaign per season because it was feasible but also sufficient for our study objectives, as we did not want to use it for budgeting, but to study the pattern and variability. However, we would like to point out that these intensive 24 h campaigns required a great deal of logistical and human effort, as all available emission pathways were measured using manual and, in some cases, large chambers (including spatial replicates), and water, reed and sediment samples were collected and analyzed. Of course, more measurement days would always be great if resources and time allowed. However, in Table 1, we have summarized the characteristics of each individual 24 h campaign to present and allow comparison with other days/nights in the same season. Furthermore, there were exactly three months between the campaigns to avoid subjective selection of the measurement days/nights. Unfortunately, the winter campaign had to be postponed due to channel sediment excavation work (including local sediment deposits) in the study area, which would have affected the measurements too much.

We believe that our median $CH_4$ emissions of the ebullition pathway measured with bubble traps during the spring and summer campaigns capture the temporal variations in fluxes during a season quite well, as they are comparable to the median of the weekly collected and continuously measured $CH_4$ ebullition rates within the reed belt, separated by seasons, in Baur et al. (2024b): In the spring season in Baur et al. (2024b), the median ± standard deviation of the measured ebullition rates from March to May is $0.04 ± 0.86$ mg $CH_4$ m$^{-2}$ h$^{-1}$, while the median flux of the ebullition pathway in the spring campaign is $0.00 ± 1.00$ mg $CH_4$ m$^{-2}$ h$^{-1}$ (see Table 2). In the summer season, when the water level was above the surface (from June to mid-July), the median ebullition rate is $1.19 ± 1.35$ mg $CH_4$ m$^{-2}$ h$^{-1}$, while the median flux of the ebullition pathway in the summer campaign is $1.35 ± 3.69$ mg $CH_4$ m$^{-2}$ h$^{-1}$ (see Table 2).

*Furthermore, the authors were unable to measure all four fluxes in all four seasons, with the exception of vegetation-mediated CH4 emissions. How does the research address the need to "incorporate all emission pathways across all seasons" (Line 39) and investigate "diel patterns of each available emission pathway during all season" (Lines 53 – 54)? Please address these questions in the main text of the manuscript.*

Thank you for pointing out that we obviously did not clearly describe this issue. Due to the intra-annual dynamics of reed wetlands, the different land covers and their interfaces and emission pathways were not always available in each season (Buchsteiner et al., 2023). Of course, we can only measure each emission pathway per season that is available. For example,

in spring the water level was high, so there was no sediment-air interface. In fall, for example, the reed wetland was dry (no water level above the surface), so there was no water-air interface. However, the water level was too low to set up bubble traps before and during the winter campaign, because the reed belt was completely dry in fall and we would have destroyed the sediment surface when installing the funnels, see L153ff. Nevertheless, ebullition rates during winter are expected to be very low due to the temperature dependence (e.g. Aben et al. (2017)).

So we studied all pathways assessable during all seasons which is a methodological advancement compared to previous studies of reed wetlands (van den Berg et al., 2020; Brix et al., 2001; Jeffrey et al., 2019). Considering the dynamic nature of wetlands, we were able to measure all accessible pathways in any given season. In the revised manuscript, we will clarify this issue to rule out any misunderstanding.

**References**

Aben, R. C. H., Barros, N., van Donk, E., Frenken, T., Hilt, S., Kazanjian, G., Lamers, L. P. M., Peeters, E. T. H. M., Roelofs, J. G. M., Senerpont Domis, L. N. de, Stephan, S., Velthuis, M., van de Waal, D. B., Wik, M., Thornton, B. F., Wilkinson, J., DelSontro, T., and Kosten, S.: Cross continental increase in methane ebullition under climate change, Nature communications, 8, 1682, https://doi.org/10.1038/s41467-017-01535-y, 2017.

Baur, P. A., Maier, A., Buchsteiner, C., Zechmeister, T., and Glatzel, S.: Consequences of intense drought on CO2 and CH4 fluxes of the reed ecosystem at Lake Neusiedl, Environ. Res., 262, 119907, https://doi.org/10.1016/j.envres.2024.119907, 2024a.

Baur, P. A., Henry Pinilla, D., and Glatzel, S.: Is ebullition or diffusion more important as methane emission pathway in a shallow subsaline lake?, Sci. Total Environ., 912, 169112, https://doi.org/10.1016/j.scitotenv.2023.169112, 2024b.

Brix, H., Sorrell, B. K., and Lorenzen, B.: Are Phragmites-dominated wetlands a net source or net sink of greenhouse gases?, Aquat. Bot., 69, 313–324, https://doi.org/10.1016/S0304-3770(01)00145-0, 2001.

Buchsteiner, C., Baur, P. A., and Glatzel, S.: Spatial Analysis of Intra-Annual Reed Ecosystem Dynamics at Lake Neusiedl Using RGB Drone Imagery and Deep Learning, Remote Sens., 15, 3961, https://doi.org/10.3390/rs15163961, 2023.

Hammer, U. T. (Ed.): Saline lake ecosystems of the world, Monographiae Biologicae, 59, Junk, Dordrecht [et al.], 616 pp., 1986.

Jeffrey, L. C., Maher, D. T., Johnston, S. G., Maguire, K., Steven, A. D. L., and Tait, D. R.: Rhizosphere to the atmosphere: contrasting methane pathways, fluxes, and geochemical drivers across the terrestrial–aquatic wetland boundary, Biogeosciences, 16, 1799–1815, https://doi.org/10.5194/bg-16-1799-2019, 2019.

Kallingal, J. T., Lindström, J., Miller, P. A., Rinne, J., Raivonen, M., and Scholze, M.: Optimising CH4 simulations from the LPJ-GUESS model v4.1 using an adaptive Markov chain Monte Carlo algorithm, Geosci. Model Dev., 17, 2299–2324, https://doi.org/10.5194/gmd-17-2299-2024, 2024.

Riley, W. J., Subin, Z. M., Lawrence, D. M., Swenson, S. C., Torn, M. S., Meng, L., Mahowald, N. M., and Hess, P.: Barriers to predicting changes in global terrestrial methane fluxes: analyses using CLM4Me, a methane biogeochemistry model integrated in CESM, Biogeosciences, 8, 1925–1953, https://doi.org/10.5194/bg-8-1925-2011, 2011.

van den Berg, M., van den Elzen, E., Ingwersen, J., Kosten, S., Lamers, L. P. M., and Streck, T.: Contribution of plant-induced pressurized flow to CH4 emission from a Phragmites fen, Sci. Rep., 10, 12304, https://doi.org/10.1038/s41598-020-69034-7, 2020.

Vroom, R., van den Berg, M., Pangala, S. R., van der Scheer, O. E., and Sorrell, B. K.:
Physiological processes affecting methane transport by wetland vegetation – A review,
Aquat. Bot., 182, 103547, https://doi.org/10.1016/j.aquabot.2022.103547, 2022.

---

## Author Response (AR1)

**Dear editor and reviewers,**

**Hereby, we resubmit our manuscript "Pathways of CH₄ formation and emission in the subsaline reed wetland of Lake Neusiedl" with a new title "Temporal dynamics of CH₄ emission pathways in the subsaline reed wetland of Lake Neusiedl", after addressing the revisions suggested by the reviewers.**

**We greatly appreciate the reviewers for their time, valuable inputs and suggestions, which we believe have significantly improved our manuscript. We are confident that this new version is suitable for readers of EGU Biogeosciences.**

**In the following, the reviewer's comments are set in *italic* and the authors' responses are marked in blue. All references to line in this document pertain to the manuscript where these changes are tracked.**

**Authors' response to Referee 1**

*Review of "Pathways of CH4 formation and emission in the subsaline reed wetland of Lake Neusiedl"*

*Here, Baur et al. continued the extensive research previously conducted by this scientific group in a reed wetland located at a lake in Austria, the title argue that they were focusing on understanding CH4 formation pathways. Although the title attempts to encompass this complex objective, I believe the methodology employed does not fully address such broad scenarios. Instead, I recommend emphasizing diel variations across different seasons rather than the overall CH4 formation processes in the studied ecosystem. I mention this because you are inferring CH4 pathways without performing an appropriate experimental analysis. Microbial assessments combined with isotopic measurements from emissions alone might not be sufficient to clearly identify these pathways. For robust conclusions, incubation experiments using substrates or more detailed analyses would be necessary. I am currently providing general comments on the manuscript, in the next review round, I will likely offer more detailed feedback. At this stage, however, I think the manuscript needs substantial improvement.*

We thank the reviewer for the comment and agree that the title was perhaps a bit too broad and that not everything could be fully answered with this study. However, we wanted to provide insights into the various emission pathways, seasonal isotopic source signatures, methanogenic characterization (following Whiticar et al. (1986)), and the microbial communities in the sediments involved in this ecosystem. We changed the title to "Temporal dynamics of CH₄ emission pathways in the subsaline reed wetland of Lake Neusiedl".

*Firstly, the Introduction appears poorly organized. Although I understand you might be addressing topics similar to those from previous manuscripts, here the Introduction is brief and omits important details. Regarding isotope information, please ensure that you clearly describe and contextualize isotopic methods and their relevance in terms of diel variations, and their respective emission. Additionally, having paragraphs consisting of only one sentence is inadequate, please expand or combine such sentences with relevant information to provide greater clarity. Several process definitions are merely mentioned but insufficiently explained. For example, you need to clearly introduce the effects of sulfate and oxygen availability on sediments, as well as their potential impacts on CH4 emissions. Not all variations in CH4 emissions can be explained by oxidation alone; rather, the microbial community may also*

*utilize alternative substrates for respiration. This aspect is currently missing and should be addressed explicitly in the Introduction.*

Thank you for your recommendations. We improved the Introduction chapter by expanding the explanations of the various processes in the sediments involved (L42ff) and the purpose of stable carbon isotope ratios (see L30ff). We also reorganized the chapter's structure (see L21ff). However, we must point out that this study does not examine the diel variability of isotope values (e.g. from atmospheric air); it only examines the seasonal variability of the isotopic source signature of $CH_4$ in the sediments. Therefore, this topic is not included in the introduction. We have added the section 2.2.3 "Stable carbon isotope ratios" to the Material and Methods chapter, where we describe the methods used to analyze the stable carbon isotope ratios (see L259ff). The description of measuring stable carbon isotope ratios in chamber air during field campaigns is found in L210ff, and the description of measuring them in the collected ebullition gases in the lab in L227ff.

*Secondly, I understand that the methodology was adopted from previous studies, however, you need to briefly include specific details about your own methods. For example:*

*(i) Please include, either as an appendix or supplementary material, information about the observed flux determined using the chambers and the Picarro analyzer, clearly indicating instances when data were too erratic. Additionally, specify the Picarro measurement mode and the frequency of data acquisition. For isotope analyses, clarify how long measurements were stabilized before recording values, as well as the measurement uncertainties (I remember it is necessary like 6 min to confirm the isotopic value).*

Yes, these details are mentioned in Baur et al. (2024b), but for better traceability, we have added more information in the Material and Methods chapter. The Picarro measurement mode and Picarro measurement frequency during the field chamber measurements have been added in L214f. The ebullition gases were measured with the Picarro and a small sample isotope module in the lab shortly after the campaign. The measurement mode and duration (8 minutes) were added in L232ff. In addition, we added the section 2.2.3 "Stable carbon isotope ratios" to provide a better description of the isotope analyses (see L259ff). The uncertainty of the stable carbon isotope ratios during measurements are added in L223f and L233f. The uncertainties of the isotopic source signatures are presented as error bars in Figure 3.

Flux determination is described in section 2.2.2 "Flux calculation" (see L244ff) and covers two cases in which the chamber measurement data from the quality-checked $CH_4$ fluxes were excluded: Firstly, if ebullition occurred during the closure of the chamber types intended for measuring diffusion-related fluxes. Second, if the $R^2$ value of the linear regression of the dry $CH_4$ mole fraction was less than 0.6 and the $CH_4$ flux was not minimal (close to zero) and the $R^2$ value of the dry $CO_2$ mole fraction was also less than 0.6, as this would indicate insufficient performance of the chamber measurement, which was almost never the case. Almost all exclusions occurred during the summer campaign due to the very sensitive and soft sediment layer (and the very low water level), which artificially triggered ebullitions during chamber closure times. Overall, however, 213 $CH_4$ chamber fluxes could be used and showed a mean $R^2$ value of 0.9. In addition, 72 ebullition rates were calculated from the bubble traps. Data such as $R^2$, slope, temperature, and the quality-checked flux value of each chamber measurement were published in the data file in the repository. This makes it possible to trace which measurements received an NA (-9999) for the flux value and were therefore excluded from further analysis.

*(ii) Table 1 currently provides limited information; please add the sample sizes corresponding to the reported uncertainties. Lastly, regarding the laboratory-analyzed samples (although not clearly described), could you clarify if the reported standard deviations for sulfate, TOC, and chlorophyll-a resulted from temporal sampling conducted over a 24-hour period?*

Yes, most parameters in Table 1 show the temporal means and standard deviation over 24 h, but the table also includes 3 spatial replicates (see the 3 water sample locations in Figure 1d). Only for two parameters (water level and leaf area index), which are not assumed to change over 24 h, only spatial means and standard deviation are given. We have made this clearer by including this information in the caption of Table 1 and also adding the sample sizes. This table is located in section 2.2 "Measurement setup" of the Material and Methods chapter and describes the environmental conditions at the site during each 24 h campaign. The intention of this table was also to show and allow their comparability with other days/nights in the same season (L168f). In order to keep the manuscript concise, we refrained from including the water sample data in the results chapter in the initial submission. However, as the reviewer's question indicates that this might be of great interest to the reader, we have included the diel variability of the water parameters in the Appendix (see Figure A2) and added the water analysis methods in the section 2.3 "Water sampling and analysis" (see L280ff).

*Also, I am curious about how you obtained ebullition data during periods of very low water levels, particularly in summer when this emission pathway appears most prominent in your results.*

In the Baur et al. (2024b) study, the bubble traps were permanently installed from March to July 2021 until the water level in the reed belt was no longer above the surface. As the traps were installed at a time of higher water levels (early March 2021), we were able to measure ebullition (collection of trapped gases) with these traps at a time of very low water levels in early summer (June/July). The information about the installation time was added in L242f. However, for our study here, the gas collection interval of the bubble traps was shorter (every 6 h) to investigate if there was a diel pattern, to quantify the ebullition pathway during 24 h and also to compare it with the other assessable emission pathways. For the rest of the time, the trapped gas was collected once a week and used for the study of Baur et al. (2024b). These results show an increase in the ebullition rate in the reed belt from March to July. If we understand your comment correctly, we do not consider the ebullition pathway to be the most important emission pathway in summer, but rather the plant-mediated transport to be the dominant emission pathway due to the maximum and highest mean flux rates (see Table 2 and Figure 2). However, the ebullition pathway showed significantly higher $CH_4$ emission rates in summer than in spring.

*Furthermore, I am surprised by the high chlorophyll-a values observed in winter, are you certain these measurements represent chlorophyll-a and not turbidity caused by shallow water levels?*

In the reed belt, the surface water has a brownish color, but it is clear and not turbid (unlike the open lake water of Lake Neusiedl). The relatively high chlorophyll-a concentrations in winter are probably more likely due to algae and very low water levels (higher concentration per liter). According to Wolfram et al. (2015), the reed belt of Lake Neusiedl is mostly inhabited by small phytoplankton species, such as cryptophytes and green algae.

*(iii) Ebullition measurements using bubble traps appear highly biased due to the short deployment duration. Given that ebullition is a very stochastic process, it is possible that*

*significant emissions were missed simply because the traps were placed away from emission hotspots. I understand the difficulty involved, as this is one of the major challenges when studying ebullition. In previous studies, you deployed bubble traps for longer periods, however, the focus on diel variations in this study limited that possibility. Do you think a single measurement day per season provides sufficient data to establish a representative ebullition pattern? Looking into the data I would not discard as the main source of emission.*

Since we had 9 bubble traps (always 3 traps on one board) collecting bubbles in the reed belt, we believe that the spatial heterogeneity of ebullition and ebullition hotspots are well reflected. In order to investigate the diel pattern in different seasons in this study, the gas collection time of the bubble trap had to be shorter. Combined with the other study on ebullition rates at different sites and over a longer period of time (Baur et al., 2024b), we have a better picture of ebullition at Lake Neusiedl. Of course, it is always better to have more measurement days, but since 24 h campaigns are very resource intensive, we were not able to conduct more such campaigns. In addition, the drying out of the reed belt was not foreseen, otherwise longer measurements of the bubble traps would have been possible (after July).

*The soil chambers faced several issues related to ebullition (as you commented in the method section), suggesting that significant amounts of CH4 accumulated in the surface sediments, which may have been missed due to the short deployment time of the bubble traps (in this case absence of water, but still emission). Given this limitation, why did you not consider using the ebullition events detected by the Picarro analyzer for fluxes to determine other sources of CH4? With proper data management, you could potentially quantify ebullition as part of the total CH4 flux using these measurements.*

Yes, in the summer campaign there were ebullitions at the beginning or during chamber closure in 8 out of 24 sediment chamber measurements. These ebullitions were likely triggered by the movement of the chamber closure, because the top sediment layer ("fluffy layer") was very mobile, as mentioned in the Material and Methods chapter in L248ff. Thus, we consider this release not representative. Since one aim of the study was to investigate the diel pattern of the naturally released methane emissions for each assessable pathway and season (and not the total ecosystem emissions, which were investigated in Baur et al. (2024a)), we refrained from using the possibly artificially triggered ebullitions at the beginning or during the sediment chamber closure. As the experimental design has foreseen to measure the diffusion pathway with sediment chambers and the ebullition pathway at the same time (in summer) with the bubble traps, we did not want to overestimate this pathway by double counting. Since the bubble traps collected a lot of bubbles during the summer campaign, we believe that we are taking the accumulation of methane in the top sediment layer into account, at least to a certain extent.

*Thirdly, the presented results represent measurements performed over just 1 day per season. I understand this limitation, it is because you were also collecting data on additional topics, which have been integrated into other published manuscripts. Although you made a substantial effort to collect this data, some results may still be significantly biased due to the short sampling period, particularly for ebullition as previously mentioned.*

Yes, only one 24 h campaign was conducted per season. This may lead to some limitations, but we also had to consider the feasibility and the purpose of this study. We still believe that this study contributes to understand the diel variations in this ecosystem and its emission pathways, despite only one year of measurements. In addition, we have published a study with continuous eddy covariance measurements that provide information on the total methane fluxes of this ecosystem over 4.5 years (see Baur et al. (2024a)). As mentioned in L475ff, it makes sense to

extend the ebullition gas collection time intervals, especially in spring. However, since higher ebullition rates are associated with higher temperatures, e.g. in summer, shorter measurement time intervals of 6 h are also useful, as the measured values show (67% of the ebullition rates greater than zero, see L358f). Since the focus of this study is on the different emission pathways, we wanted to compare the ebullition rate with the other emission rates and their diel pattern.

*But also, plant-mediated emissions are not fully isolated, as total emissions originating from sources external to the plants could have contributed.*

We are not sure if we understood your comment the way you meant. Here are our responses on how the emissions from the vegetation chambers or the plant-mediated transport itself might have been affected:

We agree that for vegetation chambers, the plant-mediated transport of reeds within the chambers cannot be completely isolated from diffusion from the underlying sediment or water surface. However, the major contribution or dominance of plant-mediated transport in the emission rates in the vegetation chamber compared to the diffusion pathway (at the water-air or sediment-air interfaces) is quite clear, as can be clearly seen from the much higher emission rates from the vegetation chambers in each season compared to the sediment or floating chambers. Therefore, we believe that the effect on the total emissions in the vegetation chambers should be relatively small. Furthermore, vegetation chambers are a common method for estimating plant-mediated transport of reed plants, see also van den Berg et al. (2020) or Jeffrey et al. (2019).

We also agree that the reed stems are probably well connected to each other by their rhizomes and are most likely all form "one" plant. However, this makes it almost impossible to isolate one reed stem by any measures (e.g. clipping, etc.) without destroying the reed ecosystem in that area. We had the opportunity to conduct research in the core zone of a national park with an exceptional permit and were therefore instructed to disturb the ecosystem as little as possible. We kept the chamber closure time as short as possible to minimize artefacts. Due to the installation of the bottom frame of the chambers in advance (in the winter before the campaigns), we had to cut the sediment at least 7 cm deep (see L199), which of course also meant that the rhizome connections with the surrounding area might have been severed.

*The figures require additional information, as it is currently difficult to interpret the origin of the presented data. For instance, in Figure 2, it's unclear what each data point represent, are these individual measurements or average? If they represent average values, please clearly indicate this and the uncertainties. Additionally, if a polynomial function has been applied, specify its order. However, I question whether using a polynomial fit is appropriate in this context; please clearly justify its application as a fit rather than presenting the raw temporal data directly. Also what is the point to fit to this polynomial?*

In Figure 2, each point represents a single flux rate. We made this clear in the caption and added the order of the polynomial function (3$^{rd}$). Since the data points represent the raw flux time series, we used the polynomial fit (with the 95% confidence interval) to visualize the average (and uncertainty) diel pattern of each emission pathway, including the peak (if available). An advantage of a polynomial fit is that it can also show smaller temporal variations, and in our case, it gave us the opportunity to show the temporal pattern with the raw data points and the exact measurement time without aggregating over time (e.g. one measurement round). We consider this advantageous as for example, in summer we had to measure 3 different manual chamber types with 3 spatial replicates each, it took quite some time to measure all of them one

after the other (including flushing time of the cable and the measuring cell of the gas analyzer) and therefore the light or other meteorological conditions could have changed during one measurement round.

*Regarding Figure 3, the source of the data is unclear, and I am particularly interested in understanding why these results differ substantially from those previously published in your earlier manuscript in Baur et al. (2024, DOI: 10.1016/j.scitotenv.2023.169112). There is no clear oxidation pattern in Figure C. So you need to expand the information about it. Does this figure include all isotopic signature data from Figures A and B in the four points? Additionally, could you clarify at what value is used during the Picarro measurements?*

The data points shown in Figures 3a and b (Keeling plots) are in the x axis the reciprocal of the mean dry mole fraction of $CH_4$ (or $CO_2$) and in the y axis the mean stable carbon isotope ratio $\delta^{13}C$ of $CH_4$ (or $CO_2$), measured with the Picarro G2201-i during the closure of each chamber measurement. These points are used to calculate the seasonal isotopic source signatures (y-intercept of the regression line) using the Keeling plot approach. This approach is explained in the Material and Methods chapter in L260ff. However, we clarified this with a better description in the caption of Figure 3.

The methanogenic characterization plot after Whiticar et al. (1986) in Figure 3c uses the four calculated seasonal isotopic source signature pairs, which were determined in Figures 3a and b as the y-intercept values of the regression lines and their standard errors. The winter source signature with its uncertainty (standard error) colored in black in Figure 3c is partly in the range of methane oxidation. This means that in this plot the carbon fractionation factor $\alpha_c$ is between 1.005 and 1.03. To facilitate understanding of the values used in Figure 3c, we adjusted the explanation in the caption.

In Baur et al. (2024b), in contrast to the manuscript submitted here, only directly measured $\delta^{13}C$-$CH_4$ values of the collected gas from the bubble traps are used in the methanogenic characterization plot. Ebullition is known to have no isotopic carbon fractionation, unlike other processes such as diffusion or methane oxidation (see 267ff). Therefore, the isotopic values of the ebullition gases can be used directly as an indicator of the methane source in the sediments without calculating the source signatures with a Keeling plot, as it was done here for all chamber measurements. Our study confirms the results of the methanogenic characterization plot (according to Whiticar et al. (1986)) from the previous study that acetoclastic activity is indicated as the main pathway of methanogenesis in this reed ecosystem, not only for the months of March to July (Baur et al., 2024b), but for all four seasons (see Figure 3c). In addition, this study was the first to determine the seasonal source signatures for $CH_4$ and $CO_2$ for all four seasons in a reed wetland (y-intercepts in Figures 3a and b).

We clarified these points in the caption of Figure 3, but have also added section 2.2.3 "Stable carbon isotope ratios" in the Material and Methods chapter to better describe the Keeling plot approach (see L259ff).

**Authors' response to Referee 2**

*The authors assessed four pathways of CH4 fluxes (vegetation-mediated, soil- and water-air diffusion, and ebullition), pathways of methanogenesis (acetoclastic, hydrogenotrophic, and methyltrophic), microbial communities, and sediment physical and chemical properties in a subsaline reed-dominated temperate wetland. CH4 fluxes were measured over 24-hour periods 4 times over one year covering all four seasons (spring, summer, fall, winter). Only vegetation-mediated CH4 flux was measured in all 4 seasons.*

Yes, that is correct. Due to the intra-annual dynamics in this wetland with reed, the available land cover (water areas, open sediment areas, and reed stands) and thus the corresponding interfaces and emission pathways are not always available in every season (Baur et al., 2024a). For example, in spring, the water level was high, so there was no sediment-air interface. In fall, however, the reed wetland was dry (no water level above the surface), so there was no water-air interface.

*The abstract implies that the research results contribute to reducing uncertainty in estimates of CH4 emissions from wetlands and improving wetland greenhouse gas modeling. However, the link between the research results and their significance needs clarification in the main text. Specifically, how can the results improve models, and what important knowledge gaps do they fill?*

In the Abstract (see L3ff) and Introduction chapter (see L82ff), we improved and expanded the description of the knowledge gaps in the currently applied wetland $CH_4$ emission models, as well as how our field study can contribute to filling these gaps.

The $CH_4$ emissions in process-based vegetation models, e.g. like LPJ-GUESS in Kallingal et al. (2024) or CLM4Me model in Riley et al. (2011), only partially implement the plant-mediated transport by considering only the passive mechanism (concentration gradient) (Riley et al., 2011; Kallingal et al., 2024). They also do not consider the pressurized flow that occurs in plant species such as *Phragmites* or *Typha* sp. and others, although it is known that, for example, the $CH_4$ emissions by plant-mediated transport of *Phragmites australis* can be more than five times higher than by diffusion (Brix et al., 2001). According to Vroom et al. (2022), most other wetland greenhouse gas models do not implement plant-mediated $CH_4$ fluxes in wetlands and do not include them in the total flux due to high variability in their contribution and the lack of data about this variability.

➢ Therefore, our study can help to fill important knowledge gaps
  o by quantifying the temporal contribution (seasonal and diel) of plant-mediated transport of reed wetlands and highlighting the dominance of plant-mediated transport not only in summer (as previous studies did, e.g. van den Berg et al. (2020)).
  o by better understanding these emission pathways and their temporal (diel) pattern of the studied reed wetland for each season
  o by using the findings to improve the parametrization and implementation of emission pathways (especially the plant-mediated transport of wetland plants) in the models.
➢ Furthermore, the study contributes to alleviating the general lack of plant-mediated $CH_4$ flux data in wetlands, so that plant-mediated transport and especially the pressurized flow mechanism will hopefully be implemented in (more) wetland $CH_4$ emission models to reduce their uncertainties, but also to make them more realistic.

- In addition, our $CH_4$ flux data (for each assessable emission pathway) can be used to optimize wetland $CH_4$ emission models as validation data for modeled $CH_4$ emissions for plant-mediated transport, ebullition, and diffusion in process-based vegetation models.
- Moreover, the findings of this study contribute to a more comprehensive understanding of the characteristics of reed wetlands, including the temporal $CH_4$ emission dynamics, their sediment and water properties, and their microbial communities.

*Discussion of the results of sediment physical and chemical property analysis is absent and should be added.*

We have added the section 4.3 "Sediment properties of a subsaline reed wetland" in the Discussion chapter (L510ff), in which we will discuss the results of the physiochemical analysis of the sediments in the reed wetland, their seasonal changes and differences compared to other (reed) wetlands.

*The authors should also define the term "subsaline" and clarify the significance of the wetland type studied. This is particularly important for convincing non-specialists of the research impact.*

According to Hammer (1986), subsaline lakes are defined as lakes with a salinity between 0.5 and 3‰. We have added this salinity range before the cited reference in L109.

The significance of the wetland type studied can be demonstrated for several reasons, which we clarified in the Introduction chapter (see L106ff):

The studied wetland type is one of the largest connected reed wetlands in Europe and has a special water characteristic for inland waters, e.g. subsalinity. Subsaline wetland ecosystems, and especially subsaline reed ecosystems, are rarely ever studied and therefore need to be (further) investigated. Because of the salinity, these ecosystems are to some degree more comparable to brackish coastal ecosystems than to freshwater ecosystems. However, the salt composition of the reed wetland in this study differs from that of coastal salt marshes because the main salt is sodium bicarbonate, not sodium chloride. It also differs greatly from other inland ecosystems such as reed fens (see e.g. van den Berg et al. (2020)) due to different soil properties. In addition, the studied lake, in which more than half is covered by the reed wetland, has been severely affected by droughts since mid-2015, which have influenced the carbon cycle, its processes and fluxes (Baur et al., 2024a). The international importance of this wetland is also shown by its cross-border protected areas as a Ramsar site, a UNESCO World Heritage site, and a National Park. All these made it very interesting to study this wetland type for a better understanding of carbon-related processes in different wetland types and conditions.

*I also have some concerns regarding the research design using punctual 24-hour assessments of fluxes over four seasons, which may not adequately capture within-season temporal variation in fluxes, particularly ebullition. Further justification of 24-hour monitoring periods and comparison of flux pathways across seasons is needed. Is there potential for bias in the seasonal measurements given the short duration of monitoring periods?*

*If the authors are willing to address these issues, I'm happy to review a revised version of the manuscript in further detail.*

We have addressed your questions in the revised manuscript:

We have taken great care to come up with meaningful results by studying the seasonal differences in the diel variation of the $CH_4$ fluxes for each assessable emission pathway of a reed wetland. We used the research design of one intensive 24 h campaign per season because it was feasible but also sufficient for our study objectives, as we did not want to use it for budgeting, but to study the pattern and variability. However, we would like to point out that these intensive 24 h campaigns required a great deal of logistical and human effort, as all assessable emission pathways were measured using manual and, in some cases, large chambers (including spatial replicates), and water, reed and sediment samples were collected and analyzed. Of course, more measurement days would always be great if resources and time allowed. However, in Table 1, we have summarized the characteristics of each individual 24 h campaign to present and allow comparison with other days/nights in the same season (see L168f). Furthermore, there were exactly three months between the campaigns to avoid subjective selection of the measurement days/nights (see L162f). Unfortunately, the winter campaign had to be postponed due to channel sediment excavation work (including local sediment deposits) in the study area, which would have affected the measurements too much (see L163ff).

We believe that our median $CH_4$ emissions of the ebullition pathway measured with bubble traps during the spring and summer campaigns capture the temporal variations in fluxes during a season quite well, as they are comparable to the median of the weekly collected and continuously measured $CH_4$ ebullition rates within the reed belt, separated by seasons, in Baur et al. (2024b): In the spring season in Baur et al. (2024b), the median ± standard deviation of the measured ebullition rates from March to May is $0.04 \pm 0.86$ mg $CH_4$ m$^{-2}$ h$^{-1}$, while the median flux of the ebullition pathway in the spring campaign is $0.00 \pm 1.00$ mg $CH_4$ m$^{-2}$ h$^{-1}$ (see Table 2). In the summer season, when the water level was above the surface (from June to mid-July), the median ebullition rate is $1.19 \pm 1.35$ mg $CH_4$ m$^{-2}$ h$^{-1}$, while the median flux of the ebullition pathway in the summer campaign is $1.35 \pm 3.69$ mg $CH_4$ m$^{-2}$ h$^{-1}$ (see Table 2). We have added these sentences in the Discussion chapter in L479ff.

*Furthermore, the authors were unable to measure all four fluxes in all four seasons, with the exception of vegetation-mediated CH4 emissions. How does the research address the need to "incorporate all emission pathways across all seasons" (Line 39) and investigate "diel patterns of each available emission pathway during all season" (Lines 53 – 54)? Please address these questions in the main text of the manuscript.*

Thank you for pointing out that we obviously did not clearly describe this issue. Due to the intra-annual dynamics of reed wetlands, the different land covers and their interfaces and emission pathways were not always available in each season (Baur et al., 2024a). Of course, we can only measure each emission pathway per season that is assessable. For example, in spring the water level was high, so there was no sediment-air interface. In fall, for example, the reed wetland was dry (no water level above the surface), so there was no water-air interface. However, the water level was too low to set up bubble traps before and during the winter campaign, because the reed belt was completely dry in fall and we would have destroyed the sediment surface when installing the funnels, see L239ff. Nevertheless, ebullition rates during winter are expected to be very low due to the temperature dependence (e.g. Aben et al. (2017)), see L485f. Since the traps were installed at a time of higher water levels in early March 2021, we were able to use these traps to measure ebullition (collection of trapped gases) at a time of very low water levels in the summer campaign, before the study site dried out (see L242ff)

So we studied all pathways assessable during all seasons which is a methodological advancement compared to previous studies of reed wetlands (van den Berg et al., 2020; Brix et

al., 2001; Jeffrey et al., 2019). Considering the dynamic nature of wetlands, we were able to measure all accessible pathways in any given season. We have clarified this issue to rule out any misunderstanding in L7, L65f, L70, L75f, L104f, L155, L157f, L438.

**References**

Aben, R. C. H., Barros, N., van Donk, E., Frenken, T., Hilt, S., Kazanjian, G., Lamers, L. P. M., Peeters, E. T. H. M., Roelofs, J. G. M., Senerpont Domis, L. N. de, Stephan, S., Velthuis, M., van de Waal, D. B., Wik, M., Thornton, B. F., Wilkinson, J., DelSontro, T., and Kosten, S.: Cross continental increase in methane ebullition under climate change, Nature communications, 8, 1682, https://doi.org/10.1038/s41467-017-01535-y, 2017.

Baur, P. A., Maier, A., Buchsteiner, C., Zechmeister, T., and Glatzel, S.: Consequences of intense drought on $CO_2$ and $CH_4$ fluxes of the reed ecosystem at Lake Neusiedl, Environ. Res., 262, 119907, https://doi.org/10.1016/j.envres.2024.119907, 2024a.

Baur, P. A., Henry Pinilla, D., and Glatzel, S.: Is ebullition or diffusion more important as methane emission pathway in a shallow subsaline lake?, Sci. Total Environ., 912, 169112, https://doi.org/10.1016/j.scitotenv.2023.169112, 2024b.

Brix, H., Sorrell, B. K., and Lorenzen, B.: Are Phragmites-dominated wetlands a net source or net sink of greenhouse gases?, Aquat. Bot., 69, 313–324, https://doi.org/10.1016/S0304-3770(01)00145-0, 2001.

Hammer, U. T. (Ed.): Saline lake ecosystems of the world, Monographiae Biologicae, 59, Junk, Dordrecht [et al.], 616 pp., 1986.

Jeffrey, L. C., Maher, D. T., Johnston, S. G., Maguire, K., Steven, A. D. L., and Tait, D. R.: Rhizosphere to the atmosphere: contrasting methane pathways, fluxes, and geochemical drivers across the terrestrial–aquatic wetland boundary, Biogeosciences, 16, 1799–1815, https://doi.org/10.5194/bg-16-1799-2019, 2019.

Kallingal, J. T., Lindström, J., Miller, P. A., Rinne, J., Raivonen, M., and Scholze, M.: Optimising $CH_4$ simulations from the LPJ-GUESS model v4.1 using an adaptive Markov chain Monte Carlo algorithm, Geosci. Model Dev., 17, 2299–2324, https://doi.org/10.5194/gmd-17-2299-2024, 2024.

Riley, W. J., Subin, Z. M., Lawrence, D. M., Swenson, S. C., Torn, M. S., Meng, L., Mahowald, N. M., and Hess, P.: Barriers to predicting changes in global terrestrial methane fluxes: analyses using CLM4Me, a methane biogeochemistry model integrated in CESM, Biogeosciences, 8, 1925–1953, https://doi.org/10.5194/bg-8-1925-2011, 2011.

van den Berg, M., van den Elzen, E., Ingwersen, J., Kosten, S., Lamers, L. P. M., and Streck, T.: Contribution of plant-induced pressurized flow to $CH_4$ emission from a Phragmites fen, Sci. Rep., 10, 12304, https://doi.org/10.1038/s41598-020-69034-7, 2020.

Vroom, R., van den Berg, M., Pangala, S. R., van der Scheer, O. E., and Sorrell, B. K.: Physiological processes affecting methane transport by wetland vegetation – A review, Aquat. Bot., 182, 103547, https://doi.org/10.1016/j.aquabot.2022.103547, 2022.

Whiticar, M., Faber, E., and Schoell, M.: Biogenic methane formation in marine and freshwater environments: $CO_2$ reduction vs. acetate fermentation—Isotope evidence, Geochim. Cosmochim. Acta, 50, 693–709, https://doi.org/10.1016/0016-7037(86)90346-7, 1986.

Wolfram, G., Großschartner, M., and Krisa, H.: Algen – Plankton – Fische: Der Neusiedler See aus limnologischer Sicht, Arbeitsgemeinschaft Natürliche Ressourcen, Eisenstadt, Austria, 48 pp., 2015.